# Indoor air surveillance and factors associated with respiratory pathogen detection in community settings in Belgium

Joren Raymenants [1,2] ✉, Caspar Geenen [1], Lore Budts[3], Jonathan Thibaut [1], Marijn Thijssen[4], Hannelore De Mulder[1], Sarah Gorissen[1], Bastiaan Craessaerts[3], Lies Laenen [1,3], Kurt Beuselinck[3], Sien Ombelet[3], Els Keyaerts[3,5] & Emmanuel André[1,3,5]

Currently, the real-life impact of indoor climate, human behaviour, ventilation and air filtration on respiratory pathogen detection and concentration are poorly understood. This hinders the interpretability of bioaerosol quantification in indoor air to surveil respiratory pathogens and transmission risk. We tested 341 indoor air samples from 21 community settings in Belgium for 29 respiratory pathogens using qPCR. On average, 3.9 pathogens were positive per sample and 85.3% of samples tested positive for at least one. Pathogen detection and concentration varied significantly by pathogen, month, and age group in generalised linear (mixed) models and generalised estimating equations. High $CO_2$ and low natural ventilation were independent risk factors for detection. The odds ratio for detection was 1.09 (95% CI 1.03–1.15) per 100 parts per million (ppm) increase in $CO_2$, and 0.88 (95% CI 0.80–0.97) per stepwise increase in natural ventilation (on a Likert scale). $CO_2$ concentration and portable air filtration were independently associated with pathogen concentration. Each 100ppm increase in $CO_2$ was associated with a qPCR Ct value decrease of 0.08 (95% CI −0.12 to −0.04), and portable air filtration with a 0.58 (95% CI 0.25–0.91) increase. The effects of occupancy, sampling duration, mask wearing, vocalisation, temperature, humidity and mechanical ventilation were not significant. Our results support the importance of ventilation and air filtration to reduce transmission.

Many respiratory infections are transmitted via the airborne route[1–7]. Airborne transmission is almost exclusively an indoor phenomenon[3,8–10]. Its risk to susceptible attendants depends on pathogen, host, behavioural and environmental/building related factors[9,10]. Pathogens differ in their ability to colonise hosts and survive in the environment while retaining infectiousness[11,12]. The number of hosts, their respiratory activity, mask wearing and individual predisposition influence aerosol generation[4,13–16]. Environmental/building related factors such as room volume and airflow patterns, temperature, humidity, UV radiation, ventilation and air filtration may impact aerosol transport, settling, inactivation and removal[9–11].

There is some evidence supporting the use of ventilation to reduce infectious disease incidence. High $CO_2$ concentration, which reflects poor ventilation, was directly associated with school absence

[1]Laboratory of Clinical Microbiology, KU Leuven, Herestraat 49, 3000 Leuven, Belgium. [2]Department of General Internal Medicine, University Hospitals Leuven, Herestraat 49, 3000 Leuven, Belgium. [3]Department of Laboratory Medicine, National Reference Center of Respiratory Pathogens, University Hospitals Leuven, Herestraat 49, 3000 Leuven, Belgium. [4]Laboratory of Clinical and Epidemiological Virology (Rega Institute), KU Leuven, Herestraat 49, 3000 Leuven, Belgium. [5]These authors contributed equally: Els Keyaerts, Emmanuel André. ✉e-mail: joren.raymenants@kuleuven.be

due to illness and with common cold symptoms[15,17]. Low air exchange rates per person through mechanical ventilation were associated with higher incidence of pneumococcal disease during a prison outbreak and with a higher risk of tuberculin conversion in healthcare workers[4,18]. The evidence to support transmission reduction by means of portable air filters—which are more affordable than pathogen removal by classical Heating, Ventilation and Air Conditioning (HVAC) systems[19]—is more limited. They were associated with a reduced incidence of invasive aspergillosis and reduced surface contamination with *Methicillin-resistant Staphylococcus Aureus*[20,21].

The quantification of respiratory pathogens or their genetic material in indoor air has been used to study the influence of environmental factors on disease transmission. This approach has the advantage of not requiring clinical follow-up of attendants. Dinoi et al. (2022) recently combined data from 73 studies performing qPCR on indoor air samples and showed that the SARS-CoV-2 bioaerosol load was lowest in outdoor air, and higher in indoor air from hospitals than from community settings, which points at the link between pathogen detection, by qPCR, and the risk of transmission for occupants[22]. In other studies, indoor $CO_2$ concentration was associated with higher detection of rhinovirus bioaerosols in ambient air, higher concentration of bacterial cell wall components and culturable bacterial colony forming units[23,24]. The presence of an advanced mechanical ventilation system with high-efficiency particulate absorbing (HEPA) filtration, directional flow or increased air changes per hour (ACH), correlated with lower fungal colony forming units per unit volume in hospital settings[25]. On the other hand, bacterial bioaerosol loads were similar across areas with mechanical, advanced mechanical and natural ventilation in the same study. Another recent study reported that severe acute respiratory syndrome coronavirus 2 (SARS-CoV-2) viral copies were more abundant in aerosols collected in closer proximity to an infected individual placed in a controlled environment. They also correlated positively with nasopharyngeal viral copies and ambient $CO_2$. On the other hand, they correlated inversely with ventilation, portable air filtration and increased humidity[26]. As for portable air filtration, experiments using particle counters showed that portable filters sped up the clearance of airborne particles[27,28]. Two small studies also suggested a reduction in detection of SARS-CoV-2 genetic material in ambient air, but the effect was not significant[29,30]. In contrast, Conway-Morris et al. (2022) did see a significant reduction in the detection, by qPCR, of SARS-CoV-2 and other respiratory pathogens[31]. No study has thus far performed a multivariate analysis which controlled for other important variables (e.g occupancy, seasonality, mask wearing, etc.) when assessing the influence of either ventilation or portable air filters on the load of respiratory pathogen bioaerosols in real-life settings.

In addition to quantifying transmission risk, sampling and testing of indoor air for respiratory pathogen bioaerosols may become an important add-on to other data sources for epidemiological surveillance, such as clinical samples, sentinel surveillance and sewage monitoring[32,33]. During the COVID-19 pandemic, pathogen detection in sewage was scaled and provided important policy insights. One benefit of environmental samples is their independence from clinical test indications, tendency for testing or laboratory capacity. Sewage sampling can surveil the population of entire cities, but also has disadvantages. Samples are highly contaminated with environmental microorganisms, runoff times may be long and variable, and—especially for respiratory pathogens—the relationship between gastrointestinal shedding and the risk of transmission may be complex[32,34]. Air sampling may be an interesting and complimentary alternative.

QPCR on ambient air has long demonstrated its ability to detect pathogen presence, concentration, and genotype[3,22,33,35,36]. A recent study demonstrated the promise of multiplex qPCR on indoor air samples from community settings to track the presence of SARS-CoV-2 and other respiratory pathogens[33]. However, before this approach can

be rolled out at scale, the factors influencing pathogen detection and concentration need better characterisation.

We aimed to empirically identify the host, pathogen, behavioural and environmental/building factors which correlate with a higher respiratory pathogen bioaerosol load, as assessed by qPCR, in indoor ambient air. We hypothesised that factors shown or suspected to contribute to airborne transmission would be associated with higher bioaerosol loads. If so, this would validate the use of qPCR on air samples as a proxy to quantify transmission risk and the effect of transmission reduction efforts. Also, these same factors would need consideration when performing qPCR on indoor air samples for epidemiological surveillance.

In a prospective study over a 7-month period, we therefore tested indoor ambient air from community settings in Belgium for 29 respiratory pathogens using qPCR. We investigated which of the following pathogen, host, behavioural and environmental/building related variables influenced their detection and concentration: the number of attendees, attendee density (number of attendees divided by room volume), sampling duration, mask wearing, vocalisation (voice use), natural ventilation (opening of doors and windows), portable air filtration, presence of mechanical ventilation, local COVID-19 incidence, indoor $CO_2$ concentration, temperature and relative humidity. See Methods for detailed definitions of each assessed variable. In an interventional sub-study, we evaluated the effect of portable air filters in a nursery. In an exploratory analysis, we investigated whether the pathogens found in ambient air samples from community settings corresponded to the pathogens found in patients with severe respiratory infections in the same region and period. We therefore retrieved the results of the same, 29 respiratory pathogen qPCR panel, performed on respiratory samples of patients at University Hospitals Leuven[37].

## Results

### Pathogen detection varies with season and age of attendants

We collected 341 environmental air samples in 21 sampling sites between October 2021 and April 2022. See Supplementary Table 1 for sampling site characteristics. Sampling durations (mean of 133 min and median of 126 min) corresponded well with the 120 min target. Two samples had missing results of the respiratory pathogen panel, while 36 had a missing result of the TaqPath SARS-CoV-2 assay (Thermo Fisher Scientific, Waltham, MA). The number of missing values for all variables is listed in Supplementary Table 1. Procedures for inferring them are described in Supplementary Methods.

When comparing positivity rates of all air samples, the most frequently detected pathogens, in descending order, were *Streptococcus pneumoniae* (58%), human enterovirus (incl. rhinovirus) (54%), human bocavirus (45%), human adenovirus (40%) and human cytomegalovirus (38%). The percentage of samples which were positive for at least one pathogen was highest in the 3- to 6-year-old age group (30/30, 100%) followed by 0-3 years (122/123, 99%), 25–65 years (9/10, 90%), 12–18 years (19/24, 79%), 18–25 years (44/57, 77%), 6–12 years (21/29, 72%) and over 65 years (46/68, 68%). Supplementary Fig. 1 shows a detailed picture of the detected pathogens by age group and time-period. As Supplementary Table 1 shows, location-specific positivity rates for at least one pathogen varied from 10 to 100%, with high variation within age categories.

Temporal variations in the positivity rates of pathogens are apparent in Fig. 1. This figure shows results from the nursery setting, which was the most stable of age groups regarding sampling frequency, occupancy and the specific individuals present in the sampling locations. Human bocavirus, human cytomegalovirus, human enterovirus (incl. rhinovirus) and *Streptococcus pneumoniae* were almost always positive. We observed a long peak of human adenovirus and *Pneumocystis jirovecii* over the winter. Other pathogens had shorter peaks, such as *Human coronavirus 229E, Human coronavirus HKU-1, Human coronavirus OC43*, enterovirus D68, influenza A virus, human

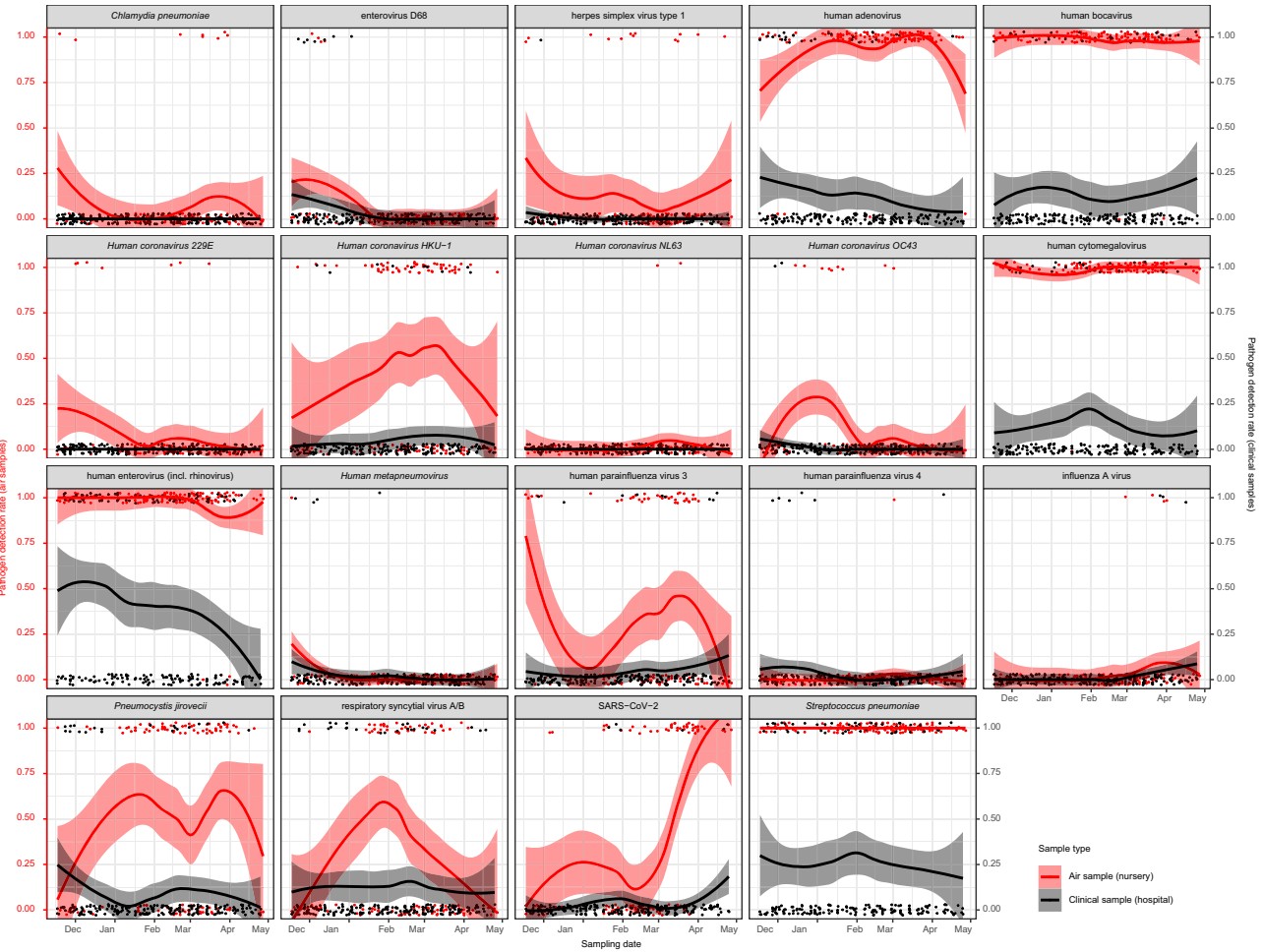

**Fig. 1 | Positivity rates of respiratory pathogens in ambient air in nursery locations (red) compared to clinical samples from a local hospital (black/grey).** Each panel represents qPCR test results of one of the 29 targeted pathogens, plotted by sampling date. Each pathogen which was positive in at least one air sample is shown. For SARS-CoV-2, the TaqPath results are shown. The individual red datapoints represent the positive and negative ambient air samples taken in a nursery (0 = negative, 1 = positive). This was the most stable age group regarding sampling frequency, occupancy and the continued presence of the same group of individuals (Supplementary Fig. 1). Red lines and shaded areas show a corresponding LOcally weEighted Scatterplot Smoothing (LOESS) regression of the positivity rate for each pathogen with 95% confidence intervals. We included 121 air samples. For comparison, we retrieved the 206 results of the same 29 pathogen multiplex qPCR respiratory panel, performed in 0–3 year old children with respiratory infections at University Hospitals Leuven between October 2021 and May 2022. This hospital is adjacent to the nursery. Individual black datapoints represent the respiratory samples (0 = negative, 1 = positive). Black lines and shaded areas show a corresponding LOESS regression of the positivity rate for each pathogen with 95% confidence intervals. An association between results from both sample types can be observed for SARS-CoV-2 (N positive air samples = 46) and, with much less positive samples, for enterovirus D68 and influenza A virus (N positive air samples = 4 for each). Supplementary Fig. 2 shows the corresponding results for all sites.

parainfluenza virus 3, respiratory syncytial virus A/B and SARS-CoV-2. Supplementary Fig. 2 shows the corresponding results for all sites.

In an exploratory analysis, we assessed whether pathogen detection rates in ambient air in community settings corresponded with those detected in patients with severe respiratory infections in University Hospitals Leuven. This hospital is adjacent to the nursery and drains most patients in the region. In the nursery, the visual association was observed most clearly for SARS-CoV-2 (N positive air samples = 46) and, with much less positive samples, for enterovirus D68 and influenza A virus (N positive air samples = 4 for each) (Fig. 1). When comparing pathogen detection rates across all sampling sites and age groups, a visual association was apparent for SARS-CoV-2 only (Supplementary Fig. 2).

## The influence of pathogen, host, behavioural and environmental/building related factors on indoor bioaerosol load
We determined independent effects of a range of variables on airborne pathogen detection and concentration, by considering the qPCR result of each pathogen in a sample as a separate observation. The pathogen was considered a covariate in the resulting models, several of which included corrections for within-sample correlation. Missing data were imputed, and for each model, backward elimination was performed until only statistically significant variables remained (*p*-value < 0.05). Subsequently, observations with imputed variables were removed to confirm the observed associations.

We excluded pathogens with less than 10 positive qPCR tests after grouping them—to increase statistical power—as follows: human parainfluenza virus 1 to 4 under 'parainfluenza viruses'; *Human coronaviruses 229E, HKU-1, NL63* and *OC43* under 'other coronaviruses'. At least 10 positive results were present for 14 pathogens before grouping and for 12 pathogens after.

## Factors associated with pathogen detection
First, positivity for any respiratory pathogen was the binary outcome in a logistic regression model. Supplementary Table 3 lists the *p*-values and odds ratios before backward elimination. Backward elimination on

**Table 1 | lists the pathogen, host, behavioural and environmental/building related factors significantly associated with indoor air bioaerosol load after backward elimination in (logistic)generalised linear models**

| Remaining variables | p-value | Adjusted odds ratio and 95% CI |
|---|---|---|
| *(a) Pathogen detection (all pathogens) in a logistic regression model* | | |
| Pathogen | <0.0001 | |
| Age group | <0.0001 | |
| Month | 0.0024 | |
| $CO_2$ | 0.0015 | 1.09 (CI 1.03–1.15) per 100 ppm increase in $CO_2$ |
| Natural ventilation | 0.0097 | 0.88 (CI 0.80–0.97) per step increase (Likert scale) |
| **Remaining variables** | **p-value** | **Coefficient and 95% CI** |
| *(b) Pathogen concentration (qPCR Ct of positive samples, all pathogens) in a linear regression model* | | |
| Pathogen | <0.0001 | |
| Age group | <0.0001 | |
| Month | 0.0020 | |
| $CO_2$ | <0.0001 | –0.08 (CI –0.12 to –0.04) per increase of 100 ppm |
| Portable air filtration | 0.0005 | 0.58 (CI 0.25–0.91) |

Panel a lists the factors significantly associated with pathogen detection in a logistic regression model. It also shows effect sizes (odds ratios and 95% CI) for $CO_2$ and natural ventilation, after adjustment for pathogen, age group and month. See Supplementary Table 4 for unadjusted estimates. Panel b) lists the factors significantly associated with pathogen concentration (measured in qPCR Ct values) in a linear regression model. It also shows effect sizes (change in Ct value and 95% CI) for $CO_2$ and portable air filtration, after adjustment for pathogen, age group and month. See Supplementary Table 4 for unadjusted estimates. *P*-values are two-sided. They were estimated using the Chi squared method (no adjustment for multiple comparisons). Almost identical results of alternative models are shown in Supplementary Table 4.

this data, which included imputed datapoints, left pathogen, month, age group, natural ventilation, $CO_2$ and vocalisation as significant variables. Unexpectedly, increased vocalisation (on a Likert scale) was associated with decreased pathogen detection. However, after exclusion of observations with imputed variables from the resulting model, vocalisation was removed as significant variable (Table 1, panel a). The odds ratio of pathogen detection was 1.09 (95% CI 1.03 to 1.15) per 100 parts per million (ppm) increase in $CO_2$ concentration. In addition, the odds ratio of pathogen detection was 0.88 (95% CI 0.80 to 0.97) per stepwise increase in natural ventilation (Likert scale). Significance levels and effect sizes were almost identical in the mixed effects logistic regression and generalised estimating equations models, both correcting for within-sample correlation (Supplementary Table 4).

To assess whether these associations held true for pathogens individually, we used the retained independent variables from these models to run a logistic regression model with backward elimination for each pathogen. These analyses had less power due to lower sample sizes. However, a significant association remained between mean $CO_2$ and detection of human enterovirus (incl. rhinovirus), other coronaviruses, *Pneumocystis jirovecii* and *Streptococcus pneumoniae*. Contradictorily, we found a negative association with the detection of human bocavirus. As for natural ventilation, it was negatively associated with the detection of *Pneumocystis jirovecii* and respiratory syncytial virus A/B. Supplementary Table 5 lists all model outcomes. Supplementary Fig. 3 shows the univariate correlations of $CO_2$ and natural ventilation with pathogen detection.

### Factors associated with pathogen concentration
Here, pathogen concentration, measured by qPCR Ct value, was the numeric outcome in a linear regression model. Supplementary Table 3 lists the *p*-values and effect sizes before backward elimination.

Backward elimination on this data, which included imputed datapoints, left pathogen, month, age group, $CO_2$ and air filtration as significant variables. Each 100ppm increase in $CO_2$ concentration was associated with a decreased qPCR Ct value of 0.08 (95% CI 0.04 to 0.12). Natural ventilation was not significantly associated with pathogen concentration, which contrasts with the previous analysis. However, air filtration was significantly associated with pathogen concentration, with a 0.58 (95% CI 0.25 to 0.91) increase in Ct value in its presence. Significance levels and effect sizes were almost identical when excluding imputed values, or when running a linear mixed effects model correcting for within-sample correlation (Supplementary Table 4, panel b).

Starting with the retained independent variables from this model, we then ran a linear regression model with backward elimination for each individual pathogen, again taking qPCR Ct values as numeric outcome. Mean $CO_2$ remained positively associated with a higher concentration (lower Ct value) of human adenovirus, human bocavirus, human cytomegalovirus, and *Streptococcus pneumoniae*. Contradictorily, it was associated with a lower concentration (higher Ct value) of respiratory syncytial virus A/B. Air filtration was associated with lower concentrations of human bocavirus, human cytomegalovirus, other coronaviruses, and *Streptococcus pneumoniae* (See Supplementary Table 6 for all model outcomes).

### Portable air filtration reduced pathogen detection and concentration in an interventional comparison
Starting from February 7th, air samples were taken simultaneously at three locations in the nursery, for 8 consecutive weeks, 3 times per week (Mondays, Wednesdays, and Fridays). Location 1 had no air filtration, location 2 had three Blue PURE 221 filters (Blueair®) installed, with a total theoretical clean air delivery rate of 1770 $m^3$/h and a resulting number of ACH of 10.7. Location 3 had three Philips 3000i filters (Philips®) installed, with a total theoretical clean air delivery rate of 999 $m^3$/h and a resulting number of ACH of 6.1 (Supplementary Fig. 4 and Supplementary Table 7).

First, we compared the positivity for any respiratory pathogen between three phases of air filtration in each location separately: no ongoing filtration (Mondays), 48 h of continuous filtration (Wednesdays) and 96 h of continuous filtration (Fridays) (Fig. 2). Cochran's Q test showed no significant difference between Mondays, Wednesdays, and Fridays in location 1 ($p = 0.6762$). In location 2, a significant difference was present across days ($p = 0.0006$). Pairwise comparisons demonstrated a difference between Mondays and Wednesdays ($p = 0.0229$) and Mondays and Fridays ($p = 0.0009$) but not between Wednesdays and Fridays ($p = 1$). In location 3, the difference between days did not reach significance, but there was a trend ($p = 0.0701$).

Next, we used linear mixed effects regression models to evaluate the change in average concentration of respiratory pathogens on Wednesdays and Fridays, compared to baseline on Mondays, in each location separately. We saw no significant change in average Ct values throughout filtration phases in location 1 ($p = 0.9506$ when comparing Mondays to Wednesdays and 0.7101 when comparing Mondays to Fridays). In location 2, there was a significant increase in Ct value of 1.22 (95% CI 0.65–1.79, $p < 0.0001$) from Mondays to Wednesdays, and 1.13 from Mondays to Fridays (95% CI 0.57–1.70, $p = 0.0002$). In location 3, the difference in Ct value was not significant when comparing Mondays and Wednesdays (Ct +0.33, 95% CI −0.32 to 0.98, $p = 0.3146$). However, there was a significant increase on Fridays compared to Mondays (Ct +1.02, 95% CI 0.37–1.67, $p = 0.0026$). Supplementary Table 8 lists all model outcomes.

### Discussion
Both the recent study by Ramuta et al. (2022) and ours demonstrate the scalability of performing multi-pathogen qPCR on air samples from community settings to highlight pathogen presence[33]. The age of

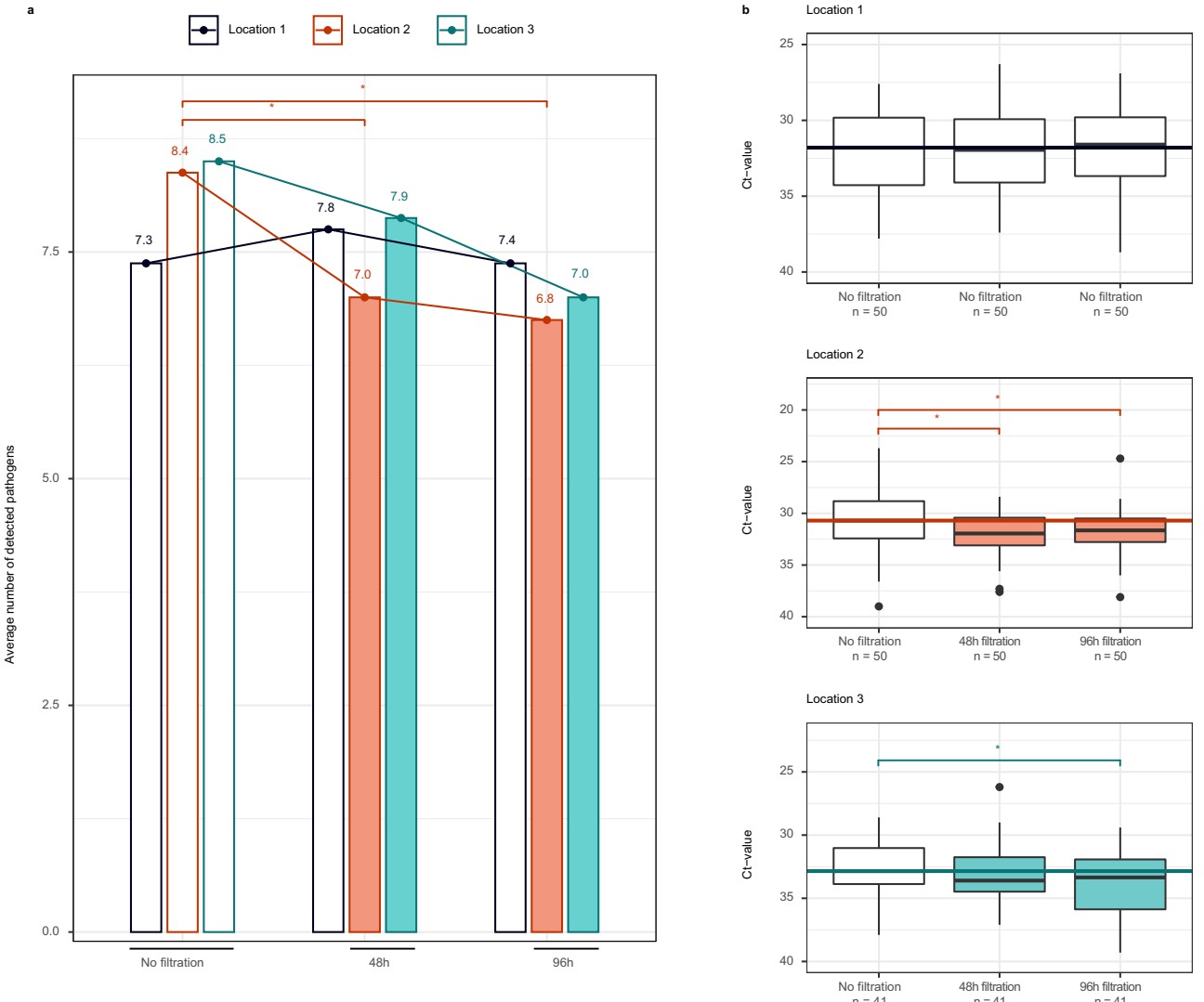

**Fig. 2 | The influence of portable air filtration on respiratory pathogen detection and concentration in ambient air.** For 8 consecutive weeks, samples were taken in three nursery locations on Mondays ($n = 8$), Wednesdays ($n = 8$), and Fridays ($n = 8$). Location 1 = control group (no air filtration); Location 2 = air filtration at 10.7 air changes per hour (ACH) starting Mondays (after sampling) and ending Fridays (after sampling); Location 3 = air filtration at 6.1 ACH starting Mondays (after sampling) and ending Fridays (after sampling). No one was present over the weekends. Panel (**a**) shows the mean number of pathogens detected in each of the nursery locations in the absence of filtration (not shaded), after 48 h of filtration (Locations 2 and 3, shaded) and after 96 h of filtration (Locations 2 and 3, shaded). * indicates a significant difference. We used a Cochran's Q to compare the three filtration phases in each location separately, followed by pairwise Cochran's Q tests when a significant difference (two-sided *P*-value < 0.05) was found, with Holm correction for multiple testing. In location 1, there was no significant difference between Mondays, Wednesdays, and Fridays ($p = 0.6762$). In location 2, a significant difference was present across days (0.0006), between Mondays and Wednesdays (pairwise comparison: $p = 0.0229$), and between Mondays and Fridays (pairwise comparison: $p = 0.0009$), but not between Wednesdays and Fridays (pairwise comparison: $p = 1$). In location 3, the difference between days did not reach

significance, but there was a trend ($p = 0.0701$). Panel (**b**) shows the evolution of Ct values of n positive pathogens (throughout the 8 weeks) in each of the nursery locations in the absence of filtration (not shaded), after 48 h of filtration (shaded) and after 96 h of filtration (shaded). Ct values for a particular pathogen were included only if the pathogen was detected in samples from all three filtration phases during the same week in one location. The horizontal line corresponds to the mean Ct value on Mondays. The central boxplot line corresponds to the median, and the lower and upper boxplot bounds to the 25th and 75th percentiles. The upper/lower whisker extend from the upper/lower boxplot bound to the largest/lowest value no further than 1.5 * IQR. Data beyond the whiskers are outliers, plotted individually. * indicates a significant difference in a linear mixed effects regression model, including week and pathogen as random effects. 95% CI were calculated using the confint command in R, and *p*-values with the Kenward–Roger approximation of the t-distribution. Ct values did not differ significantly in location 1 ($p = 0.9506$ between Mondays and Wednesdays; 0.7101 between Mondays and Fridays). In location 2, they were significantly different between Mondays and Wednesdays ($p < 0.0001$), and Mondays and Fridays ($p = 0.0002$). In location 3, they were not significantly different between Mondays and Wednesdays ($p = 0.3146$), but were between Mondays and Fridays ($p = 0.0026$).

attendants appears to be a key determinant of the type and number of detected pathogens. In both studies, the number of detected pathogens was highest in sites populated by young children. This corresponds to the incidence rate of respiratory infections across age groups and—for pathogens such as human bocavirus, human cytomegalovirus and *Streptococcus pneumoniae*—the occurrence of

prolonged shedding from the respiratory tract of young chidren[38–41]. Several pathogens, such as human adenovirus, *Pneumocystis jirovecii, Human coronavirus 229E, Human coronavirus HKU-1, Human coronavirus OC43*, enterovirus D68, influenza A virus, human parainfluenza virus 3, respiratory syncytial virus A/B and SARS-CoV-2, showed clear temporal variations in their detection rates, suggesting changing

incidence rates throughout the study (Fig. 1 and Supplementary Fig. 2). As did Ramuta et al. (2022), we observed a visual association between the detection of SARS-CoV-2 in ambient air from community settings and in clinical samples from patients in the same geographical area (Fig. 1, Supplementary Fig. 2). The same association was not readily seen for most other pathogens. As clinical respiratory panels were only performed on samples of severely ill patients in our study (see *Methods*), this may imply that the link between variations in community circulation and morbidity was stronger for SARS-CoV-2 than for other pathogens.

The current and previous studies clearly indicate that (multiplex) qPCR on ambient air from community settings can be a complementary surveillance tool to track the circulation of respiratory pathogens. However, in addition to local epidemiology, periodical differences in pathogen detection and quantity can be explained by variations in behaviour, environmental factors, technical/analytical factors, or a combination. This underscores the need to characterise the sampling sites and standardise air sampling and analysis, to interpret the epidemiological relevance of pathogen presence and concentrations in ambient air.

By testing more samples and pathogens than previous studies, we were able to show for the first time that both respiratory pathogen genomic material presence and concentration, as assessed by qPCR, were positively associated with $CO_2$ concentration, after correcting for a range of confounding variables. Results were consistent across models (logistic regression, mixed effect logistic regression, generalised estimating equations, linear regression, and linear mixed effect). Natural ventilation was also negatively associated with pathogen detection, even though our models corrected for $CO_2$ concentration. This may result from the fact that $CO_2$ is an imperfect marker for ventilation[10]. These results confirm that bioaerosol load in indoor ambient air correlates strongly with low levels of ventilation (see Table 1 and Supplementary Tables 4, 9). Pathogen specific models were generally consistent with these results, even if statistical power was more limited (Supplementary Tables 5, 6). Two exceptions were human bocavirus and respiratory syncytial virus A/B. In the former, $CO_2$ correlated negatively with pathogen detection, although positively with pathogen concentration. In the latter, natural ventilation correlated negatively with detection as expected, but $CO_2$ was negatively correlated with concentration. Type I errors or uncorrected confounders may explain these inconsistencies. The strength of the correlation between the $CO_2$ concentration and the presence of a particular pathogen was often mirrored in the strength of the inverse correlation between natural ventilation and detection of the same pathogen (Supplementary Fig. 3).

In several multivariate models, the presence of air filtration remained independently associated with a lower concentration of respiratory pathogens, measured in qPCR Ct values, even after controlling for natural ventilation and $CO_2$ concentration (Table 1, Supplementary Tables 6, 9). When analysing positivity rates in the two nursery sites with air filters, we saw a significant reduction in the number of detected pathogens during filtration in the location equipped with the highest filtration capacity (theoretical ACH of 10.7). The concentration of positive pathogens was also significantly reduced. Ct values increased by 1.13 on average (95% CI 0.57–1.70) between Mondays, when filtration had been inactive for 3 days, and Fridays, after 4 days of continuous filtration. In the location equipped with less filtration capacity (theoretical ACH of 6.1), we saw a trend towards a reduction in the number of detected pathogens. Here, the pathogen concentration was reduced significantly after four days of continuous filtration, but not after two. On Fridays, the Ct values increased by 1.02 on average (95% CI 0.37–1.67) compared to Mondays. We saw no difference in the control group (Location 1) (Fig. 2, Supplementary Table 8). The observed effect and dose–response relationship confirm the efficacy of air filtration to reduce the respiratory pathogen bioaerosol load, given sufficient filtration capacity.

The current study demonstrates that the type of pathogen, seasonality, ventilation and air filtration influence the respiratory pathogen bioaerosol load. However, many questions remain before multiplex qPCR on ambient air samples can become a new standard to surveil the community circulation of respiratory pathogens.

Firstly, technical aspects related to sampling need consideration. The type of sampler and its flow rate, sampling duration and the volume of the sampled room may all influence pathogen detection. We did not compare air samplers, but did use one with a comparably high flow rate[42,43]. Within the narrow range in our study, the sampling duration was not independently associated with pathogen detection or concentration (Supplementary Table 3).

Secondly, laboratory analysis methods need to be standardised and validated. In our study, we observed a difference in SARS-CoV-2 detection rates between qPCR platforms, with the ORF1ab aimed qPCR in the respiratory panel being significantly less sensitive than the TaqPath COVID-19 assay (Supplementary Methods and Supplementary Table 10). This lower sensitivity had been observed in validation experiments on clinical samples, but did not negatively impact accuracy in routine clinical practice (Supplementary Methods and Supplementary Table 11). Using the alternative SARS-CoV-2 qPCR as input had no impact on the main multivariate regression models (Supplementary Table 9). Their statistical power may actually have been higher, had the multiplex qPCR panel been more sensitive. Low sensitivity may however be particularly important when using inherently diluted air samples for epidemiological surveillance. For our study specifically, we cannot exclude an additional effect of the different transport buffers used for the TaqPath qPCR as opposed to the respiratory panel, or longer turn-around-times for the latter, on SARS-CoV-2 detection rates. Both were however analysed in a reasonable timeframe. The median processing time was 0.92 days (range 0.26 to 14.25, IQR 0.47–1.45) for the TaqPath qPCR and 3.32 days (range 0.79 to 16.23, IQR 2.02–5.22) for the multi-pathogen respiratory panel.

Thirdly, environmental/building related factors which were either not significant or not assessed in our study, may still need consideration. The influences of temperature and humidity on bioaerosol load, which were not significant in our models, are known to be pathogen specific and often non-linear[44]. As our main analyses used either positivity or concentration of any respiratory pathogen as primary outcome to identify linear relationships, they may not have captured the importance of these variables. UV radiation was not assessed in our study, and is unlikely to influence bioaerosol loads measured by qPCR, as it neutralises the replication potential of pathogens without physically removing their genetic material from the environment[3]. The presence of an HVAC system in the sampling sites was similarly not significant in our models. This may result from the large variation in installations across sites (Supplementary Table 7) or the fact that their effect was obscured by other variables, such as $CO_2$ concentration. The absence of a significant association between occupancy and bioaerosol load could be similarly explained. Behavioural factors such as mask wearing and vocalisation were not retained in our models, even if they are known to have an important influence on aerosol generation[16]. Contradictorily, increased vocalisation was even associated with decreased pathogen detection, although it was removed as a significant variable after exclusion of observations with imputed variables. Possible reasons for both not being associated with higher bioaerosol load are a lack of power, the fact that they were the variables most often imputed in the dataset (Supplementary Table 2), and a confounder effect, as mask wearing may coincide with the implementation of other mitigation measures.

Lastly, the sensitivity of an air sample depends on its positioning and air mixing patterns in the room. Samplers were always placed off

the ground and at maximum distance from attendants to avoid sampling resuspended aerosols or large exhaled droplets rather than airborne particles (Supplementary Table 1). Resuspension of pathogens that either survive intact or whose genetic material is most stable may indeed skew environmental surveillance data based on qPCR[11]. Similarly, the concentration of airborne pathogens is known to be greater in proximity to an infectious individual[9,26,45]. This stresses the importance of distancing the sampler from attendants, perhaps even within the HVAC system, which may also allow the inclusion of more individuals per sample[43].

Our study has several limitations. First, we did not attempt to isolate replication-competent virus, relying exclusively on qPCR analysis, or collect biological samples from attendants. This limits our ability to link risk factors with the risk of transmission directly. Second, we did not determine the exact concentrations of respiratory pathogens in ambient air as no standard curves were developed for each pathogen and qPCR platform. While this does not negate the conclusions on significant variables, it does influence the transferability of effect sizes in terms of changes in qPCR Ct values to other settings not using the exact same qPCR panels. Third, natural and mechanical ventilation rates and airflows were not assessed directly or modelled comprehensively. This limits our ability to determine whether the proximity of attendants to the sampling device may have influenced bioaerosol detection and concentration. Lastly, neither our convenience sample of community settings nor the clinical samples from patients admitted for severe respiratory infections in the nearby hospital can be considered entirely representative of the locally circulating respiratory pathogens. Our study therefore did not allow to directly compare ambient air sampling, syndromic surveillance, or sentinel sampling of clinical samples at a local level.

In conclusion, these results provide strong empirical support for the use of ventilation and air filtration to reduce transmission risk, consistent with previous studies. They further demonstrate that ambient air qPCR testing can scale to surveil community circulation of respiratory pathogens, if confounders such as $CO_2$ concentration are accounted for.

## Methods
### Air sample collection
Between October 2021 and April 2022, we collected indoor ambient air in a convenience sample of community settings in and around the city of Leuven, Belgium. Sampling sites covered different predominant age groups: nursery (0–3 years), preschool (3–6 years), primary school (6–12 years), secondary school (12–18 years), adults (18+) and nursing homes (65+). See Supplementary Table 1 for detailed characteristics of the sampling sites and Supplementary Table 7 for descriptions of the HVAC systems present in six sites. We focused on children and older people because of high incidence and morbidity from respiratory infections in these populations[40]. For university auditoria, rooms where high $CO_2$ values were registered in the weeks prior to the start of the study were selected for inclusion.

We sampled for 2 h unless site-specific schedules required shorter sampling (e.g. lunch time in schools). An AerosolSense active air sampler collected air in standard AerosolSense Capture Media (Thermo Fisher Scientific, Waltham, MA) (see Supplementary Table 1 for its positioning). This is an impaction-based active air sampler with multiple nucleic acid collection media. Air was sampled at a rate of 200 L/min through a vertical collection pipe and impacted onto the collection media. The flow rate of the AerosolSense sampler is calibrated continuously by measuring the pressure drop across the nozzle and calculating the mass flow rate for orifice, and adjusted through a PID controller. We measured environmental parameters such as $CO_2$ and humidity either manually (registering the highest recorded value while holding a Testo 435-4 device at arm's length for 20 s) or using a remote climate sensor (Elsys®, placed adjacent to the air sampler at

maximum distance from attendants). We used the former for 58 samples and the latter for 283.

### Clinical sample collection
We retrieved the results of respiratory panels performed in patients at University Hospitals Leuven in the same period[37]. This hospital drains most patients in the wider Leuven region. Respiratory panels are only performed for specific clinical indications. In immunocompetent individuals, they are performed for respiratory infections that require intensive care admission or that do not respond to initial therapy. In immunocompromised patients, they are performed more readily in the presence of lower respiratory infections.

### Air sample processing and analysis
After removal of the standard AerosolSense Capture Media cartridges (Thermo Fisher Scientific, Waltham, MA) from the sampler, they were transported to the lab on the day of collection. One of two sponges was lysed in transport buffer (DNA/RNA Shield, Zymo Research) to be used for the TaqPath qPCR assay for SARS-CoV-2. The other was lysed in Universal Transport Medium (UTM), to be used for the multiplex qPCR respiratory panel. Samples were stored at 4 degrees until processing. If they required storage over the weekend, they were frozen at −80 degrees Celsius.

### Nucleic acid extraction
For the TaqPath qPCR analysis, we used the MagMAX™ Viral/Pathogen II (MVP II) Nucleic Acid Isolation Kit for automated extraction (Thermo Fisher Scientific, AM1836) on 200 µl sample input. For internal control, samples were spiked with a purified MS2 bacteriophage as per the manufacturer's instructions (Thermo Fisher Scientific, A47817). Extracted RNA was eluted from magnetic beads in 50 µl MagMAX Viral/Pathogen Elution Buffer.

For the multiplex respiratory panel, Total Nucleic Acid (TNA) extraction started from 500 µl of air sample in UTM with NucliSens extraction reagents on easyMAG or eMAG (BioMérieux, Lyon, France). We used the specific B protocol on the instrument after off-board lysis for 10 min and continuous shaking. A 10 µL mixture of Phocine Distemper Virus and Phocine Herpesvirus was added to the lysed sample before extraction as RNA and DNA internal controls[46,47]. The elution volume of TNA was 110 µl.

### Detection of SARS-CoV-2 in air samples by RT-qPCR (TaqPath)
Extracted RNA was eluted from magnetic beads in 50 µl of UltraPure DNase/RNase free distilled water. RT-qPCR testing was performed with the TaqPath COVID-19 CE-IVD RT-PCR kit (Thermo Fisher Scientific). Results were analysed using the FastFinder analysis software (Ugentec, Belgium) and expressed as a cycle threshold (Ct) for the ORF1ab, N, and S gene targets (see also Cuypers et al.[48]).

### Detection of 29 respiratory pathogens in air samples by multiplex qPCR (respiratory panel)
An in-house respiratory panel, consisting of 12 real-time multiplex qPCRs, was run in 96 well plates on QuantStudio DX (Thermo Fisher Scientific, Waltham, MA, USA). The end volume of each PCR reaction mix was 20 µL: 5 µL of TNA, 5 µL of master mix (TaqMan Fast Virus Mix, Thermo Fisher Scientific, Waltham, MA, USA) and 10 µL of primer/probe mix (Supplementary Table 12). The temperature profile used was as follows: 50 °C for 10 min followed by 20 s at 95 °C and 45 cycles of 3 s at 90 °C and 30 s at 60 °C.

The panel detects seven non-viral pathogens (*Mycoplasma pneumoniae*, *Coxiella burnettii*, *Chlamydia pneumoniae*, *Chlamydia psittaci*, *Streptococcus pneumoniae*, *Legionella pneumophila* and *Pneumocystis jirovecii*) and twenty-two viruses: influenza A virus, influenza B virus; human parainfluenza viruses 1 to 4; respiratory syncytial virus A/B; human enterovirus (incl. rhinovirus); enterovirus D68; herpes simplex

virus type 1; herpes simplex virus type 2; *Human metapneumovirus*; human adenovirus; human bocavirus; human parechovirus; *Human coronaviruses 229E, HKU-1, NL63* and *OC43*; human cytomegalovirus; Middle East respiratory syndrome coronavirus (MERS-CoV); SARS-CoV-1/2 through the ORF1ab target. Since all positive results for ORF1ab were attributed to SARS-CoV-2, rather than SARS-CoV-1, the panel could detect 22 viruses and 29 pathogens in practice. Both an RNA (Phocine Distemper Virus) and DNA internal control (Phocine Herpesvirus-1) were run with each panel[46,47]. Two internal quality control samples (Respi 3 and Respi 4) were run on alternating days with the respiratory panel. This is standard practice in clinical routine. Respi 3 contains positive material for human bocavirus, *Chlamydia psittaci, Human coronaviruses NL63, 229E* and *HKU-1*, MERS-CoV, human enterovirus (incl. rhinovirus), enterovirus D68, herpes simplex virus type 1, *Human metapneumovirus*, influenza A virus, human parechovirus and respiratory syncytial virus A/B. Respi 4 contains positive material for human adenovirus, *Chlamydia pneumoniae, Human coronavirus OC43*, SARS-CoV-1/SARS-CoV-2 Wuhan, *Coxiella burnettii*, human cytomegalovirus, herpes simplex virus type 2, influenza B virus, *Legionella pneumophila, Mycoplasma pneumoniae, Streptococcus pneumoniae*, and *Pneumocystis jirovecii*, human parainfluenza viruses 2 and 3, respiratory syncytial virus A/B.

Supplementary Table 12 lists all target genes, primer/probe sequences and final concentrations, amplicon sizes and Ct thresholds.

The specificity was validated using External Quality Control (EQC) samples, cultures and clinical samples. The analysis was performed under ISO15189:2012 accreditation. Supplementary Methods and Supplementary Tables 11, 12 and 13 provide further details on experiments conducted to validate the respiratory panel in clinical practice and the methods used to exclude non-specific amplification in air samples.

Test results were downloaded from the University Hospitals Leuven laboratory information system as CSV files.

### Detection of 29 respiratory pathogens in clinical human respiratory samples by multiplex qPCR

Clinical samples which are analysed using the multiplex qPCR respiratory panel undergo the same procedure as air samples did in our study. They are transported to the laboratory in UTM immediately after collection. Lysis, storage, nucleic acid extraction and the detection of pathogens follow identical procedures.

Test results were downloaded from the University Hospitals Leuven laboratory information system as CSV files.

### Detailed definition of host, behavioural and environmental/building related factors collected for each sample

- Weekly COVID-19 incidence Leuven: COVID-19 incidence for the city of Leuven in the seven days until the day before sampling, per 100,000 inhabitants[49].
  The following variables were registered before and after each collected air sample:
- Predominant age group at the sampling site: 0–3 years, 3–6 years, 6–12 years, 12–18 years, 18–25 years, 25–65 years, +65 years. (Supplementary Table 1).
- Month of sampling.
- Number of attendees, measured at the start and end of each sample and averaged.
- Attendee density: averaged number of attendees divided by sampling room volume (m³). The number of attendees was estimated by headcount both at the start and end of sampling and averaged per sample.
- Sampling duration: in minutes, manual entry per sample.
- Mask wearing: estimated by Likert scale (no one, almost no one, minority, majority, almost everyone, everyone) at the start and end of sampling. Average per sample.

- Vocalisation: estimated by Likert scale (no one talks, only teacher talks, minority talks, majority talks, everyone talks, singing) at the start and end of sampling. Average per sample.
- Natural ventilation: estimated by Likert scale (no natural ventilation, one window open, door open, multiple windows open, door and window open) at the start and end of sampling. Average per sample.
- Air filtration: binary (enabled, disabled), manual entry per sample.
- Mechanical ventilation: binary (absent/present), manual entry per site. See Supplementary Tables 1, 7.
- Indoor $CO_2$ concentration: numeric (parts per million/ppm). Either measured manually at the start and end of each sample and averaged or measured continuously and averaged over the total sampling duration (<11 min before start of sampling until <11 min after end of sampling).
- Indoor temperature: numeric (degrees Celsius, °C). Either measured manually at the start and end of each sample and averaged or measured continuously and averaged over the total sampling duration (<11 min before start of sampling until <11 min after end of sampling).
- Relative humidity: numeric (%). Either measured manually at the start and end of each sample and averaged or measured continuously and averaged over the total sampling duration (<11 min before start of sampling until <11 min after end of sampling).

Manual data collection took place on paper, after which it was inputted in Excel version 16.68 (Microsoft®). Continuous measurements of ambient air parameters were collected on the web based platform (Grafana, Grafana Labs®) and downloaded as CSV files.

### Portable air filters

To test the effectiveness of portable air filters to reduce bioaerosol load, we placed them in two separate locations in a nursery (Locations 2 and 3). Another separate space was the control (Location 1). The same group of up to 20 toddlers and 1 to 4 caregivers occupied each location during sampling. No one was present over the weekends. Supplementary Fig. 4 shows the placement of air filters.

The study assessed two types of air filters. The Blue PURE 221 (Blueair®) is a HEPA and carbon filter-based device with a clean air delivery rate of 590 m³/h. From January 17 onwards, three devices were present in nursery location 2. On the first 7 days of air filtration in this location, the air was sampled without filtration, filtered for several hours, then sampled again with active filtration. From February 7 onwards, three Philips 3000i (Philips®) devices were additionally placed in location 3 (Supplementary Tables 1, 7). This is another HEPA and carbon filter-based device with a clean air delivery rate of at least 333 m³/h when operated in "turbo" mode as per manufacturer specifications. The devices were used in stage 2, which corresponds to a CADR of 186.7 m³/h per device. From this moment onwards, sampling took place concurrently in all three locations. On Mondays, air filtration started after the completion of 2 h of sampling. Air filtration then continued uninterrupted for 96 h. On Wednesdays and Fridays, sampling was repeated in each location, again for 2 h per day. Air filtration was discontinued after sampling on Fridays.

### Samples inclusion and exclusion

When describing pathogen detection patterns across sampling sites, age groups, and time, we considered each of the 29 target pathogens separately.

Only the TaqPath SARS-CoV-2 qPCR was considered for SARS-CoV-2, to avoid duplication and because it is more sensitive (see Supplementary Methods, Supplementary Tables 10, 13). The TaqPath qPCR was not performed on 35/341 samples between January 3rd and 14th due to financial constraints. The TaqPath SARS-CoV-2 result was

missing for one sample and the respiratory panel for two samples due to failed transport between labs.

When analysing the influence of pathogen, host, behavioural and environmental/building related factors on bioaerosol load, we excluded pathogens with less than 10 positive qPCR results after grouping them—to increase statistical power—as follows: human parainfluenza virus 1 to 4 under 'parainfluenza viruses'; *Human coronaviruses 229E, HKU-1, NL63* and *OC43* under 'other coronaviruses'.

Supplementary Table 2 lists the missing environmental/building related and behavioural factors and how the missing data was handled.

Supplementary Methods describes the procedure for imputing the missing variables.

For filtration, all datapoints from January 17th onwards were included in the main analyses assessing the influence of pathogen, host, behavioural and environmental/building related factors on bioaerosol load. For the interventional sub-study, only samples from February 7th onwards were included.

### Statistical analysis

To assess the influence of pathogen, host, behavioural and environmental/building related variables on pathogen detection, we used a logistic regression model, a generalised estimating equations model and a mixed effects logistic regression model. Each result for a particular pathogen was one observation, while positivity was the binary outcome. Each pathogen group had equal weight. The pathogen was a variable in the models. Both the generalised estimating equations model and mixed effects logistic regression model corrected for within-sample correlation of tests.

To assess the influence of the same variables on pathogen concentration, we used a linear regression model and mixed effects linear regression model. Pathogen concentration was measured by the qPCR Ct value of a positive pathogen. Again, each test for a particular pathogen was one observation, while the pathogen was considered a covariate in the model.

After imputing missing variables, as described in Supplementary Methods, we used backward elimination (until all remaining variables reached a *p*-value of < 0.05) in all models to estimate effect sizes of the most important variables. 95% confidence intervals were computed as follows: coefficient estimate ± standard error * 1.96. We used the Wald test to estimate *p*-values in generalised estimating equations models and the Chi squared test for (mixed effects) linear and logistic regression models. *P*-values were not corrected for multiple hypothesis testing. After backward elimination, we removed observations with imputed variables to confirm the results.

In an exploratory analysis, we evaluated whether the influence of variables found to be significant in the above models differed by pathogen. We ran logistic and linear regression models with, respectively, pathogen detection and Ct value as outcomes. Models were run for each detected pathogen separately, only using the retained significant variables from the models including all pathogens.

Lastly, we evaluated the effectiveness of portable air filtration by focusing on repeated samples taken in the three nursery locations. We used a Cochran's Q test to compare pathogen detection rates between three phases of air filtration for each location separately: no ongoing filtration (Mondays), 48 h of continuous filtration (Wednesdays) and 96 h of continuous filtration (Fridays). Pairwise Cochran's Q tests followed when the difference in phases was significant. We used Holm correction for multiple testing. We used mixed effects linear regression models to evaluate the effect of different air filtration phases on pathogen concentration, including week and pathogen as random effects, for each location separately. Ct values for a particular pathogen were included only if the pathogen was detected in samples from all three filtration phases during the same week. Confidence intervals were calculated using the confint command in R, *p*-values

were obtained using the Kenward–Roger approximation of the T-distribution (pbkrtest package in R[50]).

A two-sided *p*-value of ≤0.05 was considered significant in all analyses.

### Reporting summary

Further information on research design is available in the Nature Portfolio Reporting Summary linked to this article.

## Data availability

Supplementary Information contains the Supplementary methods, Supplementary figures and Supplementary tables. All data related to pathogen presence in environmental air are added in Supplementary Data 1 and 2. Supplementary Data 1 contains grouped pathogens, while Supplementary Data 2 contains ungrouped pathogens.

## Code availability

Data analysis was performed either using R script in R versions 4.0.2/4.0.3/4.1.1 or python script in python version 3.8 specifically written for this study. The analyses performed as part of this research paper are publicly available on https://github.com/jraymenants/envir-air-sampling.

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

## Acknowledgements

The study was financed through internal KU Leuven funds. J.R. and J.T. acknowledge support of the Research Foundation Flanders (FWO, grant numbers: 1S88721N to J.R; 1130423N to J.T.). The manufacturers of the air sampling and filtration devices were not involved in the design, conduct, analysis, or manuscript writing. The AerosolSense active air samplers, AerosolSense Capture Media and Blue PURE 221 devices were purchased. The Philips 3000i devices were donated by the manufacturer.

## Author contributions

The study was conceptualised by J.R., E.K., L.B. and E.A. H.D.M., L.B., J.R., J.T. and B.C. collected the data. Analysis was performed by C.G., J.T., S.O., M.T., J.R., L.L. and K.B. J.R., C.G., S.O., J.T. and E.K. wrote the manuscript. The remaining authors (L.B., M.T., H.D.M., S.G., B.C., L.L., K.B., E.A.) reviewed the text.

## Competing interests

The authors declare no competing interests.

## Ethical approval

The study received approval from the Ethics Committee Research UZ/KU Leuven (S66518, B3222022000873). No informed consent was required from occupants of the sampled environments. Sampling took place after the management of each institution had agreed to take part in the study.
