## [Peer Review File · Nature Communications]

Indoor air surveillance and factors associated with respiratory pathogen detection in community settings in BelgiumREVIEWER COMMENTS

Reviewer #1 (Remarks to the Author):

In the article, "Natural ventilation, low CO₂, and air filtration are associated with reduced indoor air respiratory pathogens," Raymenants and colleagues tested 341 indoor air samples from 21 settings for 29 different pathogens. This study aspires to evaluate key features of air exchange on pathogen detection. The goal of that approach is admirable but not remarkable in demonstrating how air surveillance can be coupled with ventilation and air exchange measures to assess the environment's health. They provide data indicating the relationship between some measures of air exchange and pathogen detection. Still, the results are not surprising, nor is it clear how this technology could be adequately applied to a different space. The last line of the summary states, "Our results support the importance of ventilation and air filtration to reduce transmission." That statement would be expected, even without this article.

Nonetheless, this group has generated a wealth of data that should be described. As the evidence mounts for using air surveillance to detect pathogens and virus transmission, this approach should become more mainstream. Therefore, this data is valuable for an audience. Unfortunately, several components of the results, discussion, and supplementary materials are not explained particularly well. These are outlined below.

1. They state that they evaluate 29 respiratory pathogens using qPCR using their in-house assay. The validation of this type of assay is not well described. The inclusion of controls is not explained clearly. They also state that results are being shown for 29 pathogens when most figures do not show 29 pathogens. For example, Supplementary Figure 5 shows 21 pathogens, while the legend says 29 pathogens. In addition, they state that the TaqPath assay yields different data than their in-house assay, but there is no rationale for including one data type versus another. This inconsistency is a concern for data presentation. The QC for this type of in-house assay needs to be clearer and more consistent throughout the paper, especially when they explicitly state that the qPCR platform impacts pathogen detection.
2. There are errors throughout the manuscript and incorrect references to papers and figures. For example, there is no call out to Figure 2. Supplementary table 8 lists 'group', but the text says 'location.' Line 117 calls out to Supplementary Figure 2 about a picture of pathogens, but Supplementary Figure 2 shows an air filter. Line 368 references citation 26, but this is not Cuypers et al., as stated in the paragraph. In addition to these errors, some figures are hard to see. For example, Supplementary Figure 7 has part of the axis cut off, which is hard to read.
3. Figure 1 shows measures of each pathogen graphically, but the data is binary – yes or no. In addition, each dot is colored red, so the reader doesn't know which points represent the nursery or the hospital. LOESS regressions are shown to model the positivity rate and plotted on the same graph. It seems like the LOESS regressions and individual points are two different measures that don't belong on the same graph. The same concern is raised in Supplementary Figure 6.
4. In Figure 1, they comment that the hospital had different virus detection patterns compared to the nursery, but there is no discussion on why this might be the case. With a paper devoted to the relationship between air quality and the detection of respiratory viruses, it is surprising that the air exchange rates between the hospital and nursery are not factored into this discussion. Finally, in the discussion, they point out that SARS-CoV-2 and influenza A virus detection were detected in periodical variations (Figure 1), but this is not obvious for influenza A.
5. Several pathogens are prevalent but don't typically cause a health concern. This is not addressed. Describing these viruses as ways to measure successful air ventilation is appropriate, but that needs to be clarified.
6. Figure 2 is not called out from the text, so the rationale and importance of this figure are never highlighted. In 2A, there are some error bars, but which samples are compared and mentioned is

unclear. If this figure is intended to show that filtration reduces virus burden, that is unclear. Moreover, it is unclear why they are comparing independent days to each other rather than just the three sites on a single day. Are they presuming that virus is highest on a Monday and decreases during the week? If so, why? This analysis is confusing.

7. Figure 2 uses the Ct value as a measure on the y-axis for pathogen concentration in the air, but this measure seems peculiar. The qRT-PCR assay likely generated a Ct value, but that doesn't reflect the pathogen concentration in the air. It is likely that the pathogen concentration in the air was changing over time, which is not accounted for in this figure.

Shelby OConnor

Reviewer #2 (Remarks to the Author):

Raymenants et al. report findings from an air sampling surveillance strategy for 29 viral and non-viral respiratory pathogens from 21 community settings, with the primary objective of assessing the association between indoor air quality measurements, human behavior, and ventilation. The authors also conducted a sub-study evaluating the effect of portable air filtration on bioaerosol detection and concentration in a nursery. The authors found that routine air sampling, consistent with other studies, is effective in detecting evidence of respiratory pathogens in indoor community settings. They also found that high CO₂ and low natural ventilation were associated with pathogen detection and that CO₂ concentration and air filtration were associated with pathogen concentration. The authors conclude that ventilation and air filtration are important to pathogen transmission in indoor settings.

Overall, the study presents an impressively large amount of data collected as part of the surveillance approach. This data is valuable to the scientific community, but more importantly the public health community where such strategies might better inform mitigation strategies. There are some questions of clarity I have primarily regarding presentation of results, interpretation of findings, and the statistical analysis. Below are specific comments and suggestions the authors might consider to further enhance the quality of their manuscript.

General Comments

While there are different views among the scientific community on how best to communicate the pathogen positivity data from air sample collections, given that one of the limitations to using air sampling results to inform risk assessment and mitigation policies is the lack of viability data, I suggest that the authors make it more clear throughout the paper that what is being detected in the samples is genomic material and not necessarily viable pathogens. This can be indicated in the beginning of the manuscript, methods section, and/or throughout.

Summary

Would be helpful to include the risk estimates and 95% CI's for the CO₂/natural ventilation > detection and CO₂/air filtration > concentration statements to communicate more clearly the directionality and magnitude of the associations.

Introduction

First sentence: Suggest "...respiratory infections are transmitted via the airborne route".

Suggest combining first and second paragraph into a single paragraph. As written, this is a little confusing. Airborne transmission risk to individuals indoors is a combination of environmental/building, host, and pathogen factors. The current phrasing implies that the factors influencing aerosol generation and aerosol settling time are independent. For example, temperature and humidity are also important factors to aerosol generation, in addition to settling

time. Suggest reworking.

Page 4, Paragraph 1: please specify "other important variables"

Results

Page 6, Paragraph 1: Given the variation of detection that occurred across sampling sites within different age groups, I suggest including the range of detection.

Page 7, Paragraph 2: Did the authors consider using a Fisher's Exact test?

Page 7-8: Please include the risk estimates for each of the discussed models including 95% CI's. This helps clarify the magnitude and directionality of each assessed association. Univariate estimates are provided for the CO₂/natural ventilation and pathogen detection. I do not see where univariate/unadjusted estimates are included for the multivariate models. The authors should also clarify what the statistical cut-off was for backward elimination.

Page 9: it would be helpful to also include the average number of detected pathogens for each treatment group.

Page 10: I am a little confused regarding the comparison of the SARS-CoV-2 multiplex and TF assays. First, the inclusion of this, while understandable in relationship to detection sensitivity, seems incomplete without including additional information about sample preparation, concentrations thresholds, standards, etc. It also calls into question the limits of detection of the larger in-house multiplex assay. The authors address sensitivity in the discussion section, but it seems somewhat disjointed in connection with the assay comparison. I am not sure this comparison adds value to the overall paper without a more extensive description of the assay parameters and without comparing any other assays for the other respiratory pathogens.

Discussion

General Comment: The discussion section is lacking inference regarding how pathogen factors might have influenced the results. I note that several of the most frequently detected pathogens are known to be highly persistent in the environment (i.e. adenovirus, bocavirus, cytomegalovirus, etc.). These factors may result in a detection bias, particularly compared to pathogens that are more sensitive to degradation. Pathogen factors also influence detection limits in both the sampler and the downstream laboratory assays. Including discussion of this would enhance the conclusions.

Page 10, Paragraph 3: there are many studies that use outdoor air surveillance that would challenge this statement. I suggest the authors revise to focus on how the data shows how indoor air surveillance is effective for detecting human pathogens as opposed to qualifying it against outdoor air surveillance as either/or. There are contextual arguments that could support both approaches.

Page 11, Paragraph 3: Please indicate directionality of the association.

Page 13, Paragraph 2: I suggest revising the title of this section to "Sensitivity of bioaerosol sampling".

Page 14, Paragraph 1: see comments for Page 10.

Materials and Methods

Page 15: more details regarding how air sampling was conducted is needed. Where was the sampler located at each sampling site? Was the same sampler used for every site? How was the flow rate calibrated?

Page 16, Paragraph 1: Were the samples transported on ice? The range of sample processing

times are quite wide though the median is within acceptable time limits to minimize viral degradation. This suggests outliers samples that had long holding times. The authors should address this here or in the limitations section.

Page 16-17: The authors state that the specificity of the in-house assay was validated, but no data on this is presented. I did not see a reference. Is this assay published elsewhere? Seeing the multiplex validation data would strengthen the interpretability and reproducibility of the results.

Tables

Table 1: Suggest including the unadjusted ORs.

Reviewer #3 (Remarks to the Author):

A. Summary of the key results

1. 85% of air samples from a variety of community settings (schools, university sites, bar, elderly care homes) were positive by qPCR for at least one of 29 respiratory pathogens.
2. The number of pathogens detected and their concentration varied by pathogen, month, and age group.
3. Ambient CO₂ concentration (an indicator of ventilation), natural ventilation, and filtration were associated with detection or concentration of the pathogens.
4. Several other factors (occupancy, mask wearing, vocalization, temperature, humidity, and mechanical ventilation) were not significant.

B. Originality and significance

The main contribution of the study is a large data set on the presence of airborne pathogens in community settings, coupled with assessment of ventilation and filtration in the rooms.

C. Data, methodology, statistics

The clarity and thoroughness could be improved.

D. Conclusions

I am concerned that the significance of the relationships reported (e.g., significant for CO₂ and not significant for occupancy, mask wearing, etc.) is very dependent on the quality of data representing these factors. As the authors note, natural and mechanical ventilation rates were not assessed directly.

E. Clarity and context

The clarity and context could be improved. The authors should consider citing the paper "Quantifying Environmental Mitigation of Aerosol Viral Load in a Controlled Chamber With Participants Diagnosed With Coronavirus Disease 2019."

1. line 126: "For SARS-CoV-2, enterovirus D68 and influenza A virus, variations in positivity corresponded with results from clinical samples in University Hospitals Leuven, which is located adjacent to the sampling location." The data source and analysis supporting this statement should be described in more detail.
2. line 132: This paragraph would fit better in the Methods section.
3. line 147: "Contradictorily, increased vocalization was associated with decreased pathogen detection." Further explanation is needed; otherwise this result casts doubt on the rest of the findings.
4. line 151: "per stepwise increase in natural ventilation." I was unable to find a description of how natural ventilation was quantified in steps.
5. line 188: In the analysis of the relationship between filtration and positivity for any respiratory pathogen, it would be easier for the reader to interpret the results if they simply referred to days with vs. without filtration, rather than specific days of the week. Whether the day was Monday, Wednesday, or Friday should not matter. What matters is whether filtration was running or not.
6. line 223: The relevance of outdoor air sampling to this study is not clear.

7. line 343: Manual measurements of CO₂ could easily be influenced by the exhaled breath of nearby people, so careful selection of measurement location is required.
8. SI: The methods refer to a "Linkert" scale rather than the "Likert" scale.

REVIEWER COMMENTS

Reviewer #1 (Remarks to the Author):

In the article, “Natural ventilation, low CO₂, and air filtration are associated with reduced indoor air respiratory pathogens,” Raymenants and colleagues tested 341 indoor air samples from 21 settings for 29 different pathogens. This study aspires to evaluate key features of air exchange on pathogen detection. The goal of that approach is admirable but not remarkable in demonstrating how air surveillance can be coupled with ventilation and air exchange measures to assess the environment's health. They provide data indicating the relationship between some measures of air exchange and pathogen detection. Still, the results are not surprising, nor is it clear how this technology could be adequately applied to a different space. The last line of the summary states, “Our results support the importance of ventilation and air filtration to reduce transmission.” That statement would be expected, even without this article.

Nonetheless, this group has generated a wealth of data that should be described. As the evidence mounts for using air surveillance to detect pathogens and virus transmission, this approach should become more mainstream. Therefore, this data is valuable for an audience. Unfortunately, several components of the results, discussion, and supplementary materials are not explained particularly well. These are outlined below.

1. They state that they evaluate 29 respiratory pathogens using qPCR using their in-house assay. The validation of this type of assay is not well described. The inclusion of controls is not explained clearly. They also state that results are being shown for 29 pathogens when most figures do not show 29 pathogens. For example, Supplementary Figure 5 shows 21 pathogens, while the legend says 29 pathogens. In addition, they state that the TaqPath assay yields different data than their in-house assay, but there is no rationale for including one data type versus another. This inconsistency is a concern for data presentation. The QC for this type of in-house assay needs to be clearer and more consistent throughout the paper, especially when they explicitly state that the qPCR platform impacts pathogen detection.

The in-house assay we used had indeed previously not been published in peer-reviewed literature, despite it having been used in routine clinical practice since 2016 by the Belgian National Reference Centre for Respiratory Viruses, which is based in University Hospitals Leuven, and despite it being accredited by the Belgian accrediting agency (BELAC) under ISO15189:2012. As part of this accreditation, the respiratory panel is subjected to continuous and rigorous internal and external quality control procedures. We provided more details on validation experiments in a new section of *Supplementary Methods* (“Validation of the respiratory panel in clinical practice”). Also, we highlighted the highest level of validation to which each pathogen specific qPCR contained in the respiratory panel was subjected following the last change to that qPCR (*Supplementary Table 12*).

The addition of Phocine Distemper Virus and Phocine Herpesvirus-1 as RNA and DNA internal controls, respectively, was mentioned in the methods section under “nucleic acid extraction”. We now added more emphasis by adding the same information and the use of two internal quality control samples to the section thereunder: “Detection of 29 respiratory pathogens in air samples by multiplex qPCR (respiratory panel)”.

We apologise for the lack of clarity on which pathogens are visualised when and why.

The first results section, “Pathogen detection varies with season and age of attendants” considered each of the 29 targeted pathogens individually. This section refers to *Figure 1* and (mistakenly) to *Supplementary Figure 2*, which should have been *Supplementary Figure 5*. We should have also referred to *Supplementary Figure 6*.

- We noted under *Figure 1* that “Pathogens which were positive in at least one air sample are shown”. To make clear that pathogens were considered individually in this figure, we changed the phrasing to “Each of the 29 targeted pathogens which was positive in at least one air sample is shown.” and specified that “For SARS-CoV-2, the TaqPath results are shown.”
- *Supplementary Figure 5* shows an “Overview of air samples and qPCR results for 29 pathogens.”. We had indeed not specified that the criterium to show a pathogen was that it be positive in at least one air sample. We added that now.
- While we referred to *Supplementary Figure 6*, which shows the corresponding results for all sites, in the caption of *Figure 1*, we did not in the main text. We thus added the following reference to the main text: “*Supplementary Figure 6* shows the corresponding results for all sites.” We also added to the main text why we visualised the nursery results in the main article too look at variations in pathogen detection over time: “This figure shows results from the nursery setting, which was the most stable of sampling sites regarding sampling frequency, occupancy and the specific individuals present in the sampling location.”
- In this section, we referred to *Supplementary Table 1* to show that positivity rates varied significantly by sampling location, including within one age category. The column, which listed the mean number of pathogens found per sample in each site, erroneously referred to “Mean N of pathogens respiratory panel” since the TaqPath was considered for SARS-CoV-2 rather than the respiratory panel. This was now changed to “Mean N of detected pathogens”.

As we mentioned in the second paragraph of the second results section, “The influence of host, pathogen, behavioral and environmental factors on indoor bioaerosol load”, We excluded pathogens with less than 10 positive tests after grouping them – to increase statistical power – as follows: human parainfluenza virus 1 to 4 under ‘parainfluenza viruses’; *Human coronaviruses 229E, HKU-1, NL63* and *OC43* under ‘other coronaviruses’.

- This is why models assessing the influence of host, behavioural and environmental factors on the detection and concentration of individual pathogens (see *Supplementary Figure 4* and *Supplementary Tables 8* and *9*) list groups of pathogens (like “Parainfluenza viruses” or “Other coronaviruses”) instead of the individual pathogens.
- We added this distinction in descriptions of individual pathogen occurrence, versus the analysis of the influence of pathogen, host, behavioural and environmental/building related factors on bioaerosol load, to *Methods*, under “Samples inclusion and exclusion”.
- Similarly, *Supplementary Figure 7* shows pathogens or pathogen groups, as per the above method, if they were positive at least 10 times. This was made more explicit in the figure caption.

While we did highlight that we showed the TaqPath qPCR for SARS-CoV-2 in *Supplementary Figure 5*, we did not in *Figure 1* and *Supplementary Figure 6*. This has now been added to the caption of both.

Our justification for using the TaqPath qPCR was to avoid duplication, as highlighted under “Samples inclusion and exclusion” in the *Methods*. We now added to this section that we suspected lower sensitivity. We add further justification under “Investigating the influence of the qPCR panel on SARS-CoV-2 detection in ambient air samples.” In *Supplementary Methods*. We state there that the specific epidemiological importance of SARS-CoV-2 merited added attention and that we had previous experience performing TaqPath qPCRs on air samples, but not respiratory panels.

In order to further alleviate concerns that the SARS-CoV-2 qPCR panel we used may have influenced our main results shown in *Table 1*, we repeated those analyses using the alternative SARS-CoV-2 results (from the respiratory panel) as inputs instead of the TaqPath qPCR results. The results were very similar (see *Supplementary Table 7*).

2. There are errors throughout the manuscript and incorrect references to papers and figures. For example, there is no call out to *Figure 2*. *Supplementary table 8* lists ‘group’, but the text says ‘location.’ Line 117 calls out to *Supplementary Figure 2* about a picture of pathogens, but *Supplementary Figure 2* shows an air filter. Line 368 references citation 26, but this is not Cuypers et al., as stated in the paragraph. In addition to these errors, some figures are hard to see. For example, *Supplementary Figure 7* has part of the axis cut off, which is hard to read.

We apologies for these errors, which were now duly corrected.

We added a callout to *Figure 2* under the corresponding results section.

We made sure we consistently referred to nursery locations, and not nursery groups.

We changed the erroneous call out to *Supplementary Figure 2* to the correct *Supplementary Figure 5*.

We added the correct reference to *Cuypers et al (2022)*.

We made sure the axis is fully visible in *Supplementary Figure 7*.

We updated several references which were either preprints that had been published in peer reviewed journals or had minor errors in their citation layout.

3. *Figure 1* shows measures of each pathogen graphically, but the data is binary – yes or no. In addition, each dot is colored red, so the reader doesn’t know which points represent the nursery or the hospital. LOESS regressions are shown to model the positivity rate and plotted on the same graph. It seems like the LOESS regressions and individual points are two different measures that don’t belong on the same graph. The same concern is raised in *Supplementary Figure 6*.

Only individual qPCR test results from the nursery’s ambient air samples were shown as individual red datapoints in the initial *Figure 1*, and from all sampling sites in *Supplementary Figure 6*. In the updated *Figure 1*, we added the individual datapoints from the clinical samples in University Hospitals Leuven, in black. The figure captions have been updated to better describe what the individual datapoints represent. We opted not add the 1522 individual datapoints from the hospital to *Supplementary Figure 6*.

We would however argue that, if clearly described what each element represents, combining the LOESS regression with the individual datapoints they are based on in the same graph adds to the clarity of both.

We welcome further comments on the updated graphs.

4. In *Figure 1*, they comment that the hospital had different virus detection patterns compared to the nursery, but there is no discussion on why this might be the case. With a paper devoted to the relationship between air quality and the detection of respiratory viruses, it is surprising that the air exchange rates between the hospital and nursery are not factored into this discussion. Finally, in the

discussion, they point out that SARS-CoV-2 and influenza A virus detection were detected in periodical variations (Figure 1), but this is not obvious for influenza A.

As we outlined in the first results section, the title and caption of *Figure 1*, and *Methods*, the pathogen detection rates from the University Hospital were derived from clinical samples rather than ambient air samples. To make this clearer, and strengthen the rationale for this analysis, we made the following changes to the manuscript:

- In the introduction, we added a section to the last paragraph, which explains the study setup, to better explain this exploratory analysis.
- In the corresponding results section, we discussed this exploratory analysis in a separate paragraph. We added that the visual association was based on much less positive samples for enterovirus D68 and Influenza A than it was for SARS-CoV-2. In the discussion section, we now only discuss the association in specific terms for SARS-CoV-2.
- We added the LOESS regression based on the clinical sample results to *Supplementary Figure 6*, to make it equivalent to *Figure 1*. Its caption has been altered accordingly.
- Lastly, the lack of representativeness of both our convenience sample of community settings and the clinical samples from patients admitted for severe respiratory infections in the nearby hospital, to assess pathogen circulation in the community, was added to limitations.

5. Several pathogens are prevalent but don't typically cause a health concern. This is not addressed. Describing these viruses as ways to measure successful air ventilation is appropriate, but that needs to be clarified.

We agree that some of the detected pathogens only cause limited morbidity in the general public. We also agree that some of the detected pathogens can be shed for prolonged periods of time in asymptomatic hosts. Lastly, we agree that qPCR may pick up non-viable genetic material from the environment, which could skew our detection rates towards genetic material which is most stable in the environment.

These aspects were addressed in more detail in the updated discussion section.

6. Figure 2 is not called out from the text, so the rationale and importance of this figure are never highlighted. In 2A, there are some error bars, but which samples are compared and mentioned is unclear. If this figure is intended to show that filtration reduces virus burden, that is unclear. Moreover, it is unclear why they are comparing independent days to each other rather than just the three sites on a single day. Are they presuming that virus is highest on a Monday and decreases during the week? If so, why? This analysis is confusing.

We regret that *Figure 2* and the corresponding analysis did not come across clearly.

We now call out *Figure 2* in the corresponding results section.

In this interventional sub study, we did hypothesize that the number and/or concentration of respiratory pathogens would be lower on Wednesdays and Fridays, as opposed to Mondays, in the nursery locations with portable air filters present (Locations 2 and 3). The reason is that control samples were taken on Mondays in the absence of filtration, after which air filtration was started and continued until after the last air sample was taken on Fridays.

We chose to only compare pathogen detection and concentration within each nursery location, as the same individuals were present throughout the week in that one location. This, we estimated, would minimize confounders, if we used statistical tests suited for repeated sampling setups. Furthermore, we did not aim to make a head-to-head comparison between types of air filters, but rather placed two different air filters in two separate groups to increase our confidence that air filtration as an intervention was effective for reducing the respiratory pathogen bioaerosol load. The control group (Location 1) without air filters present was added to further strengthen our confidence, while also keeping a 'clean', continuous dataset on ambient air sampling in a nursery site, without intervening.

Filtration did reduce both the number and concentration of pathogens in one location (Location 2), which had the strongest filtration in terms of ACH, as highlighted with an *. The other group, where the number of ACH was lower, only had a significant reduction in the average concentration of pathogens between Mondays (no filtration) and Fridays (96 hours of continuous filtration). This was also highlighted*.

We hope to have improved the clarity of our explanations regarding this interventional sub study in the following places: the corresponding results section, the discussion, the corresponding methods section, the caption of *Figure 2* and *Figure 2* itself, the captions of *Supplementary Figure 4* and the content and caption of *Supplementary Table 2*. In *Figure 2* for example, shading was added to locations and samples taken during active air filtration.

We also note that, besides this analysis of the interventional sub study, air filtration was one of the variables in the (mixed) logistic and linear regression models and generalized estimating equations models. Its independent association with pathogen concentration is discussed in the corresponding results section, the discussion, *Table 1* and *Supplementary Tables 5,6,7* and *9*.

7. *Figure 2* uses the Ct value as a measure on the y-axis for pathogen concentration in the air, but this measure seems peculiar. The qRT-PCR assay likely generated a Ct value, but that doesn't reflect the pathogen concentration in the air. It is likely that the pathogen concentration in the air was changing over time, which is not accounted for in this figure.

As noted in the limitations section, we opted to use the qPCR Ct value as a quantitative measure of pathogen concentration since no standard curve was available for each of the pathogens in the respiratory panel to compute the ambient air pathogen concentration in gene copies/volume of air. As described in the caption of *Figure 2*, the concentration (measured in Ct values), does indeed vary significantly in several comparisons: between Mondays and Wednesdays in Location 2, between Mondays and Fridays in Location 2, between Mondays and Wednesdays in Location 3. Significant differences are highlighted with an *.

Shelby OConnor

Reviewer #2 (Remarks to the Author):

Raymenants et al. report findings from an air sampling surveillance strategy for 29 viral and non-viral respiratory pathogens from 21 community settings, with the primary objective of assessing the association between indoor air quality measurements, human behavior, and ventilation. The authors also conducted a sub-study evaluating the effect of portable air filtration on bioaerosol detection and concentration in a nursery. The authors found that routine air sampling, consistent with other studies, is effective in detecting evidence of respiratory pathogens in indoor community settings. They also found that high CO₂ and low natural ventilation were associated with pathogen detection and that CO₂ concentration and air filtration were associated with pathogen concentration. The authors conclude that ventilation and air filtration are important to pathogen transmission in indoor settings.

Overall, the study presents an impressively large amount of data collected as part of the surveillance approach. This data is valuable to the scientific community, but more importantly the public health community where such strategies might better inform mitigation strategies. There are some questions of clarity I have primarily regarding presentation of results, interpretation of findings, and the statistical analysis. Below are specific comments and suggestions the authors might consider to further enhance the quality of their manuscript.

General Comments

While there are different views among the scientific community on how best to communicate the pathogen positivity data from air sample collections, given that one of the limitations to using air sampling results to inform risk assessment and mitigation policies is the lack of viability data, I suggest that the authors make it more clear throughout the paper that what is being detected in the samples is genomic material and not necessarily viable pathogens. This can be indicated in the beginning of the manuscript, methods section, and/or throughout.

We agree with the reviewer and put extra emphasis throughout the manuscript on the fact that we measured genetic material rather than viable pathogens.

We also removed the reference to viability in the introduction, where it was previously stated that: "QPCR on ambient air has long demonstrated its ability to detect pathogen presence, concentration, viability, and genotype".

In addition, we elaborated further on the limitations of detecting genetic material rather than viable pathogens in the discussion:

- When discussing the lack of association between temperature or humidity and respiratory pathogen bioaerosol loads.
- When discussing a possible bias towards detecting pathogens whose genetic material is most stable in the environment.

Summary

Would be helpful to include the risk estimates and 95% CI's for the CO₂/natural ventilation > detection and CO₂/air filtration > concentration statements to communicate more clearly the directionality and magnitude of the associations.

We added the effect estimates to the summary text and the 95% CI of the effect estimates to the main text. We did not add the 95% CI to the summary – keeping space constraints in mind – as the text already specifies the independent significance of the variables concerned.

Introduction

First sentence: Suggest “...respiratory infections are transmitted via the airborne route”.

This was changed as suggested.

Suggest combining first and second paragraph into a single paragraph. As written, this is a little confusing. Airborne transmission risk to individuals indoors is a combination of environmental/building, host, and pathogen factors. The current phrasing implies that the factors influencing aerosol generation and aerosol settling time are independent. For example, temperature and humidity are also important factors to aerosol generation, in addition to settling time. Suggest reworking.

We have incorporated these suggestions.

Page 4, Paragraph 1: please specify “other important variables”

With this statement, we meant pathogen, host, behavioural and environmental/building related factors that may require consideration. We added a few examples between brackets.

Results

Page 6, Paragraph 1: Given the variation of detection that occurred across sampling sites within different age groups, I suggest including the range of detection.

In the main text, we refer to *Supplementary Table 1* for more information on the range of detection in the different sampling sites. The last two columns (‘Positivity rate sample’ and ‘Mean N of detected pathogens’) demonstrate the variation in pathogen detection across sites. In the response to this comment, we also added the range in detection rates between sampling sites to the main text.

Page 7, Paragraph 2: Did the authors consider using a Fisher’s Exact test?

We suppose this comment relates to the paragraph entitled “Factors associated with pathogen presence”. We aimed to assess the influence of multiple dependent variables on the outcome of interest simultaneously, and therefore opted for (mixed) logistic regression models and a generalized estimating equations model. One of our study’s main contributions is that we accounted for multiple covariates.

Page 7-8: Please include the risk estimates for each of the discussed models including 95% CI's. This helps clarify the magnitude and directionality of each assessed association. Univariate estimates are provided for the CO₂/natural ventilation and pathogen detection. I do not see where univariate/unadjusted estimates are included for the multivariate models. The authors should also clarify what the statistical cut-off was for backward elimination.

We thank the reviewer for this comment.

We added the 95% CI to the effect estimates in the main text sections discussing the association of pathogen, host, behavioural and environmental/building related variables with pathogen detection and concentration.

Please note that we changed all references to regression models to full rather than abbreviated names. Also, we changed the wording when discussing odds of pathogen detection depending on CO₂ concentrations and stepwise increases in natural ventilation, as the previously reported percentage increases were incorrectly derived from the odds.

We only provided the univariate associations between mean CO₂ or natural ventilation and the detection of specific pathogens in indoor ambient air in *Supplementary Figure 7* to demonstrate the consistency of the data, in the sense that CO₂ was generally positively associated with detection rates of pathogens, while natural ventilation was generally negatively correlated. In response to this comment, we added a new *Supplementary Table 5*. This table lists the p-values and effect sizes (odds ratios or coefficients) for all variables in the two main models (also shown in *Table 1*), but prior to rather than after backward elimination.

Supplementary Table 6 lists the effect estimates, with 95%, for the other models. We did not include them in the main manuscript as they were almost identical. We therefore changed the wording in the main text from 'similar' to 'almost identical' to emphasize this.

The statistical cut-off for backward elimination was p-value = 0.05. This was added to the first paragraph of the corresponding results section and to *Methods* section.

Page 9: it would be helpful to also include the average number of detected pathogens for each treatment group.

We opted to add labels to the corresponding graphs in *Figure 2 Panel a*, rather than add the numbers to the main text, as this was clearer.

Page 10: I am a little confused regarding the comparison of the SARS-CoV-2 multiplex and TF assays. First, the inclusion of this, while understandable in relationship to detection sensitivity, seems incomplete without including additional information about sample preparation, concentrations thresholds, standards, etc. It also calls into question the limits of detection of the larger in-house multiplex assay. The authors address sensitivity in the discussion section, but it seems somewhat disjointed in connection with the assay comparison. I am not sure this comparison adds value to the overall paper without a more extensive description of the assay parameters and without comparing any other assays for the other respiratory pathogens.

We thank the reviewer for this feedback. We agree that the sensitivity section was a bit out of place in the main text's results section and took an outsized position in the discussion section. We therefore moved the resulting table (previously *Table 2*) to Supplementary Information (now *Supplementary Table 11*) and the methodology for this analysis to *Supplementary Methods*. We also integrated the issue of qPCR sensitivity into a broader discussion of the caveats/limitations of ambient air sampling using qPCR.

The issue of validity of the in-house multiplex qPCR panel is also discussed more thoroughly in *Methods* (when discussing the internal controls) and a new section of *Supplementary Methods*: “Validation of the respiratory panel in clinical practice”, with additional information in the previously shared *Supplementary Table 12* and new *Supplementary Table 13*. Also, more information on storage, nucleic acid extraction and pathogen detection using qPCRs is added to *Methods*.

Discussion

General Comment: The discussion section is lacking inference regarding how pathogen factors might have influenced the results. I note that several of the most frequently detected pathogens are known to be highly persistent in the environment (i.e. adenovirus, bocavirus, cytomegalovirus, etc.). These factors may result in a detection bias, particularly compared to pathogens that are more sensitive to degradation. Pathogen factors also influence detection limits in both the sampler and the downstream laboratory assays. Including discussion of this would enhance the conclusions.

We agree with these comments. As previously noted, we elaborated on these aspects in the discussion. We now discuss the following issues in more depth: prolonged shedding, the association – or lack thereof – between circulating pathogens and morbidity, the influence of technical aspects of sampling and laboratory analysis on pathogen detection, and the possible bias caused by variations in environmental stability of pathogens’ genetic material.

Page 10, Paragraph 3: there are many studies that use outdoor air surveillance that would challenge this statement. I suggest the authors revise to focus on how the data shows how indoor air surveillance is effective for detecting human pathogens as opposed to qualifying it against outdoor air surveillance as either/or. There are contextual arguments that could support both approaches.

We agree with this comment and have removed the opposition of both surveillance methods.

Page 11, Paragraph 3: Please indicate directionality of the association.

We added the directionalities of associations to this paragraph and another paragraph further down, discussing air filtration.

Page 13, Paragraph 2: I suggest revising the title of this section to “Sensitivity of bioaerosol sampling”.

We removed the subtitles in the discussion section, as per the Nature Comms author guidelines. As previously noted, we integrated the discussion on sensitivity of bioaerosol sampling into a broader discussion of its caveats/limitations.

Page 14, Paragraph 1: see comments for Page 10.

See above for response.

Materials and Methods

Page 15: more details regarding how air sampling was conducted is needed. Where was the sampler located at each sampling site? Was the same sampler used for every site? How was the flow rate calibrated?

We added more details on the height at which samplers were placed in *Supplementary Table 1*. In the discussion section and the caption of *Supplementary Table 1*, we highlighted that samplers were placed off the ground and at maximum distance from attendants to avoid sampling resuspended aerosols or large exhaled particles as much as possible. Under *Supplementary Table 1*, we also noted

that one specific sampler was used in a particular location throughout the study. A sampler could be recommitted to a different location after interruption of sampling in one location. Under *Supplementary Figure 1*, which shows the AerosolSense air sampler and discusses its characteristics, we added how the flow rate is kept constant. We refer to these supplementary sources of information in the *Methods* section.

Page 16, Paragraph 1: Were the samples transported on ice? The range of sample processing times are quite wide though the median is within acceptable time limits to minimize viral degradation. This suggests outliers samples that had long holding times. The authors should address this here or in the limitations section.

We added the IQR to the turnaround times between sample collection and result reporting. We moved this information to the discussion section focusing on possible explanations for the lesser sensitivity of the one SARS-CoV-2 qPCR over the other, and the limitations of qPCR on ambient air samples in general.

Page 16-17: The authors state that the specificity of the in-house assay was validated, but no data on this is presented. I did not see a reference. Is this assay published elsewhere? Seeing the multiplex validation data would strengthen the interpretability and reproducibility of the results.

As previously noted, the issue of validity of the in-house multiplex qPCR panel is discussed more thoroughly in a new section of *Supplementary Methods: "Validation of the respiratory panel in clinical practice"*, with additional information in the previously shared *Supplementary Table 12* and new *Supplementary Table 13*. Also, more information on storage, nucleic acid extraction and pathogen detection using qPCRs (including the use of internal controls) is added to *Methods*.

Tables

Table 1: Suggest including the unadjusted ORs.

See comments above on Results/Page 7-8, where we discuss the addition of a new *Supplementary Table 5*.

Reviewer #3 (Remarks to the Author):

A. Summary of the key results

1. 85% of air samples from a variety of community settings (schools, university sites, bar, elderly care homes) were positive by qPCR for at least one of 29 respiratory pathogens.
2. The number of pathogens detected and their concentration varied by pathogen, month, and age group.
3. Ambient CO₂ concentration (an indicator of ventilation), natural ventilation, and filtration were associated with detection or concentration of the pathogens.
4. Several other factors (occupancy, mask wearing, vocalization, temperature, humidity, and mechanical ventilation) were not significant.

B. Originality and significance

The main contribution of the study is a large data set on the presence of airborne pathogens in community settings, coupled with assessment of ventilation and filtration in the rooms.

C. Data, methodology, statistics

The clarity and thoroughness could be improved.

We hope that both clarity and thoroughness have improved satisfactorily in the reworked manuscript.

D. Conclusions

I am concerned that the significance of the relationships reported (e.g., significant for CO₂ and not significant for occupancy, mask wearing, etc.) is very dependent on the quality of data representing these factors. As the authors note, natural and mechanical ventilation rates were not assessed directly.

We agree that we used very basic methods to assess ventilation: measuring the CO₂ concentration, noting whether an HVAC was present and whether doors and windows were open. We state this as a limitation. However, these methods have the advantage of being scalable without the need for advanced equipment or expertise. In contrast, more complex and technical assessments of ventilation (e.g., in situ measurements and computational fluid dynamics) could not be repeated for each air sample in the complex set of environments surveilled in this study, nor could they be scaled in future real-life implementations of the methodology.

The fact that mask wearing and vocalization were not significant may be due to missing data (see also *Supplementary Table 3*), as noted in the updated section of the discussion.

The fact that temperature and humidity were not significant may be because we use qPCR rather than assessing pathogen viability. This was also addressed under the discussion.

The fact that occupancy and HVAC presence were not significant may be explained by confounding, as they interact with measures of ventilation and filtration which were significant across models.

E. Clarity and context

The clarity and context could be improved. The authors should consider citing the paper “Quantifying Environmental Mitigation of Aerosol Viral Load in a Controlled Chamber With Participants Diagnosed With Coronavirus Disease 2019.”

We hope both clarity and context were improved satisfactorily in the reworked manuscript. The paper mentioned had already been cited and touched upon in the introduction. We added a reference to the same paper in the discussion section.

1. line 126: “For SARS-CoV-2, enterovirus D68 and influenza A virus, variations in positivity corresponded with results from clinical samples in University Hospitals Leuven, which is located adjacent to the sampling location.” The data source and analysis supporting this statement should be described in more detail.

In response to this and previous comments, we added more details on the comparisons between ambient air samples from community settings and clinical samples from the local hospital:

- In the introduction, we added a section to the last paragraph, which explains the study setup, to better explain this exploratory analysis.
- In the corresponding results section, we discussed this exploratory analysis in a separate paragraph. We added that the visual association was based on much less positive samples for enterovirus D68 and Influenza A than it was for SARS-CoV-2. In the discussion section, we now only discuss the association in specific terms for SARS-CoV-2.
- We added the LOESS regression based on the clinical sample results to *Supplementary Figure 6*, to make it equivalent to *Figure 1*. Its caption has been altered accordingly.
- Lastly, the lack of representativeness of both our convenience sample of community settings and the clinical samples from patients admitted for severe respiratory infections in the nearby hospital, to assess pathogen circulation in the community, was added to limitations.

- Under “Clinical sample collection” in the *Methods* section, we provide more info on the indications for performing a respiratory panel in patients at this hospital.

2. line 132: This paragraph would fit better in the Methods section.

We decided to provide limited information on the analytical setup in this section since most readers will not have read the methods section first.

3. line 147: “Contradictorily, increased vocalization was associated with decreased pathogen detection.” Further explanation is needed; otherwise this result casts doubt on the rest of the findings.

We suspect that this contradictory association results from the fact that vocalization was the variable with the most missing datapoints (see *Supplementary Table 3*). Imputation of these missing variables (see *Supplementary Methods*) may have amplified the effect of the subset of samples on which imputation was based. We do not believe this casts doubt on the other significant associations, as there were much less missing datapoints for other variables and as results were very consistent across the different models.

4. line 151: “per stepwise increase in natural ventilation.” I was unable to find a description of how natural ventilation was quantified in steps.

We stipulated in this line, in other places in the text and in *Table 1* that we mean a stepwise increase on the Likert scale.

To improve the readability of the manuscript, we moved the detailed definition of all dependent variables, including the quantification of natural ventilation, to the methods section of the main text instead of *Supplementary Methods*.

5. line 188: In the analysis of the relationship between filtration and positivity for any respiratory pathogen, it would be easier for the reader to interpret the results if they simply referred to days with vs. without filtration, rather than specific days of the week. Whether the day was Monday, Wednesday, or Friday should not matter. What matters is whether filtration was running or not. We showed that active air filtration was independently associated with pathogen concentration in several multivariate models.

The interventional experimental setup, however, enabled additional analyses, of which *Figure 2* and the corresponding manuscript sections are a reflection.

For example, the location with the highest CADR/ACH (Location 2) did show a reduction in the number of detected pathogens, while there was only a trend in the other Location. Also, the while this effect was probably lost in the multivariate models. Similarly, the increase in Ct values after the initiation of air filtration in Location 2 was higher than it was in Location 3. This also suggests a dose response relationship.

We do agree that the explanations of the experimental setup of this sub study lacked sufficient clarity. We hope to have improved the clarity of our explanations regarding this interventional sub study in the following places: the corresponding results section, the discussion, the corresponding methods section, the caption of *Figure 2* and *Figure 2* itself, the captions of *Supplementary Figure 4* and the content and caption of *Supplementary Table 2*. In *Figure 2* for example, shading was added to locations and samples taken during active air filtration.

6. line 223: The relevance of outdoor air sampling to this study is not clear.

In response to this comment and a comment by another reviewer, we have removed the opposition of both surveillance methods.

7. line 343: Manual measurements of CO₂ could easily be influenced by the exhaled breath of nearby people, so careful selection of measurement location is required.

We elaborated more on this in the text: “We measured environmental parameters such as CO₂ and humidity either manually (registering the highest recorded value while holding a Testo 435-4 device at arm's length for 20 seconds) or using a remote climate sensor (Elsys[®], placed adjacent to the air sampler at maximum distance from attendants) (*Supplementary Figure 3*).”

8. SI: The methods refer to a “Linkert” scale rather than the “Likert” scale.
This was corrected.

Other comments and changes to the manuscript:

We noticed a mistake in line 247, where it was stated that each increase in CO₂ by 100ppm decreased the qPCR Ct value by 0.13. The correct number was 0.08, as stated in *Table 1, Supplementary Table 5* in the corresponding results section and – in the new version – the summary.

A similar mistake was found in line 249, where it was stated that air filtration was significantly associated with pathogen concentration, with a 0.57 increase in average Ct in its presence. The correct number, was 0.58, as stated in *Table 1, Supplementary Table 5* in the corresponding results section and – in the new version – the summary.

We noticed a mistake in line 482 of the main text, where we stated that “Samples without a respiratory panel result were included”. This should have been “excluded” and was changed accordingly in the updated manuscript.

In the first version of the manuscript, we stipulated we retrieved the results of the “respiratory panels run on respiratory samples of inpatients and outpatients at University Hospitals Leuven”. While patients may have been outpatients at the time of sampling, they were also admitted immediately thereafter. We thus simplified the wording to “respiratory samples of patients at University Hospitals Leuven”.

In the previous *Supplementary Table* listing all primer probe combinations, it was stated there was no clinical cut-off for SARS-CoV-2. This was a mistake. While there was no clinical cut off before the SARS-CoV-2 pandemic, it was placed at 38.0 after it became widely known that very high Ct values were less likely to signify a recent infection and high transmissibility. This was changed in the updated table.

We changed all references to regression models to full rather than abbreviated names, for clarity.

We changed the wording when discussing odds of pathogen detection depending on CO₂ concentrations and stepwise increases in natural ventilation, as the previously reported percentage increases were incorrect.

We removed all references to air cleaning/cleaners and consistently referred to portable air filters, to differentiate from alternative methodologies for air cleaning and from HVAC systems.

We removed the subheading under discussion, as per the formatting guidelines.

We tried to be consistent in using UK English spelling.

When stating we retrieved results of respiratory panels performed on clinical samples, we removed the 'publicly available' denotation since we used age data, which was not publicly available on the website.

REVIEWERS' COMMENTS

Reviewer #1 (Remarks to the Author):

Overall, the paper by Raymenants and colleagues is much improved from the original submission. They addressed most of my comments. However, their revisions still left a few points that could be addressed to make the manuscript even clearer to the reader and would be appreciated. They are fairly minor edits and

1. Can they please place the supplementary tables and figures in the order they appear in the text?

2. In figure 2, they compared the average Ct value of all pathogens across one site. However, relative Ct values for individual pathogens are not equivalent. Did their analysis account for each pathogen independently? What if the types of pathogens changed during the week and this affected the overall Ct value? This data is provocative, but it is still an n=3 sites. Therefore, additionally commenting on individual pathogens would be appreciated.

3. In the response to reviewers, they commented on how they showed the individual dots for the hospital cases in Figure 1. These do not appear present in the figure. Can these either be shown or can they state in the legend why they are not shown?

4. The additional explanation for Figure 1 is appreciated. It is now clear that they compared clinical data (hospital) to air surveillance data (nursery). Now that the different sample types are evident, comparing nasal swab data to air surveillance data in two different settings seems a bit deceptive in how the data is shown. Could the authors provide a second y-axis? That way, one axis clearly reflects the air results and a second axis clearly reflects the nasal swab results. Additionally, they could add a legend in the lower right empty space that refers to the difference between the black and red colors. These additional notations should make it clearer to someone who looks quickly at the figure without reading the figure legend.

Reviewer #2 (Remarks to the Author):

These revisions have increased the overall quality and readability of the manuscript. I have no further comments or suggestions for change.

Reviewer #3 (Remarks to the Author):

The authors have addressed the reviewers' suggestions satisfactorily. I am an expert on viruses in the air, and I am OK with reporting airborne concentrations in terms of Ct values with a reverse scale. I have seen data depicted this way in several other papers. I have a few additional minor comments on the revised manuscript.

1. line 50: "Environmental/building related factors such as room volume and airflow patterns, temperature, humidity, UV radiation, ventilation and air filtration may impact aerosol production, settling, inactivation and removal⁹⁻¹¹." I suggest replacing "aerosol production" with "aerosol transport" or some other movement-based phenomenon in this sentence. Building factors likely have a very minor influence, at most, on aerosol production. Production, which I think of as the number and size of aerosol particles emitted by a person, depends mainly on individual/host factors. On the other hand, the movement or transport of particles is strongly influenced by airflow patterns.

2. line 135: Supplementary Table 3 is mentioned before Supplementary Table 2. Supplementary Table 4 is not mentioned but probably should be in the paragraph reporting positivity rates (line 137). Supplementary Figure 5 is the first such figure mentioned (line 143). Does the journal want tables and figures numbered in the order presented?

3. Supplementary Figures 1-3: Are these figures of commercially available devices needed? They use images from the companies, and the information in the caption simply recapitulates marketing material. Simply listing the brand and model in the Methods section should be sufficient. These figures seem like advertisements.

4. line 187: "The odds ratio of pathogen detection was 1.09 (95% CI 1.03 to 1.15) per 100 parts per million (ppm) increase in CO₂ concentration. In contrast, the odds ratio of pathogen detection was 0.89 (95% CI 0.80 to 0.97) per stepwise increase in natural ventilation (Likert scale)."

Starting the second sentence with "In contrast" makes it sound like the second result contradicts the first. In fact, we completely expect such a relationship because an increase in natural ventilation correlates with a decrease in CO₂ concentration. If the Likert scale were defined in the opposite direction, then the odds ratio of pathogen detection would be >1 per stepwise *decrease* in natural ventilation (which correlates with an increase in CO₂ concentration). This reviewer thinks that removing "In contrast" will reduce the potential for confusion.

REVIEWER COMMENTS

Reviewer #1 (Remarks to the Author):

Overall, the paper by Raymenants and colleagues is much improved from the original submission. They addressed most of my comments. However, their revisions still left a few points that could be addressed to make the manuscript even clearer to the reader and would be appreciated. They are fairly minor edits and

1. Can they please place the supplementary tables and figures in the order they appear in the text?

We changed the order as requested.

2. In figure 2, they compared the average Ct value of all pathogens across one site. However, relative Ct values for individual pathogens are not equivalent. Did their analysis account for each pathogen independently? What if the types of pathogens changed during the week and this affected the overall Ct value? This data is provocative, but it is still an n=3 sites. Therefore, additionally commenting on individual pathogens would be appreciated.

We did compare average Ct values of all pathogens detected between phases of air filtration – no ongoing filtration (Mondays), 48 hours of continuous filtration (Wednesdays) and 96 hours of continuous filtration (Fridays). We eliminated the risk of comparing mean Ct values of different combinations of pathogens across filtration phases in two ways:

1) Ct values for a particular pathogen were included for one particular week only if the pathogen was detected in samples from all three filtration phases during that same week.

2) We included week and pathogen as random effects in our mixed effects linear regression models. Both strategies are highlighted in the methods section. We added the following sentence to the caption to emphasize this fact: “Ct values for a particular pathogen were included only if the pathogen was detected in samples from all three filtration phases during the same week in one location.”

3. In the response to reviewers, they commented on how they showed the individual dots for the hospital cases in Figure 1. These do not appear present in the figure. Can these either be shown or can they state in the legend why they are not shown?

Thank you for noticing this. They were indeed not added in this figure. We added them in the updated version.

4. The additional explanation for Figure 1 is appreciated. It is now clear that they compared clinical data (hospital) to air surveillance data (nursery). Now that the different sample types are evident, comparing nasal swab data to air surveillance data in two different settings seems a bit deceptive in how the data is shown. Could the authors provide a second y-axis? That way, one axis clearly reflects the air results and a second axis clearly reflects the nasal swab results. Additionally, they could add a legend in the lower right empty space that refers to the difference between the black and red colors. These additional notations should make it clearer to someone who looks quickly at the figure without reading the figure legend.

We made the requested changes to the figure and updated the caption accordingly.

Reviewer #2 (Remarks to the Author):

These revisions have increased the overall quality and readability of the manuscript. I have no further comments or suggestions for change.

Reviewer #3 (Remarks to the Author):

The authors have addressed the reviewers' suggestions satisfactorily. I am an expert on viruses in the air, and I am OK with reporting airborne concentrations in terms of Ct values with a reverse scale. I have seen data depicted this way in several other papers. I have a few additional minor comments on the revised manuscript.

1. line 50: "Environmental/building related factors such as room volume and airflow patterns, temperature, humidity, UV radiation, ventilation and air filtration may impact aerosol production, settling, inactivation and removal^{9–11}." I suggest replacing "aerosol production" with "aerosol transport" or some other movement-based phenomenon in this sentence. Building factors likely have a very minor influence, at most, on aerosol production. Production, which I think of as the number and size of aerosol particles emitted by a person, depends mainly on individual/host factors. On the other hand, the movement or transport of particles is strongly influenced by airflow patterns.

Agreed. We made the suggested change to this sentence.

2. line 135: Supplementary Table 3 is mentioned before Supplementary Table 2. Supplementary Table 4 is not mentioned but probably should be in the paragraph reporting positivity rates (line 137). Supplementary Figure 5 is the first such figure mentioned (line 143). Does the journal want tables and figures numbered in the order presented?

We changed the order as requested.

3. Supplementary Figures 1-3: Are these figures of commercially available devices needed? They use images from the companies, and the information in the caption simply recapitulates marketing material. Simply listing the brand and model in the Methods section should be sufficient. These figures seem like advertisements.

We agree that the devices can also be looked up by the interested reader. We deleted these figures after moving the relevant information contained in the captions to the main text methods section (on the flow rate of the air sampler and its calibration and additional information on the CADR of the Philips air cleaner).

4. line 187: "The odds ratio of pathogen detection was 1.09 (95% CI 1.03 to 1.15) per 100 parts per million (ppm) increase in CO₂ concentration. In contrast, the odds ratio of pathogen detection was 0.89 (95% CI 0.80 to 0.97) per stepwise increase in natural ventilation (Likert scale)." Starting the second sentence with "In contrast" makes it sound like the second result contradicts the first. In fact, we completely expect such a relationship because an increase in natural ventilation correlates with a decrease in CO₂ concentration. If the Likert scale were defined in the opposite direction, then the odds ratio of pathogen detection would be >1 per stepwise *decrease* in natural ventilation (which correlates with an increase in CO₂ concentration). This reviewer thinks that removing "In contrast" will reduce the potential for confusion.

Agreed. We changed 'in contrast' to 'in addition'.